# Limits to the strain engineering of layered square-planar nickelate thin films

Dan Ferenc Segedin [1,11], Berit H. Goodge [2,3,11], Grace A. Pan [1], Qi Song [1], Harrison LaBollita[4], Myung-Chul Jung [4], Hesham El-Sherif [5], Spencer Doyle[1], Ari Turkiewicz [1], Nicole K. Taylor[6], Jarad A. Mason [7], Alpha T. N'Diaye [8], Hanjong Paik[9,10], Ismail El Baggari[5], Antia S. Botana[4], Lena F. Kourkoutis [2,3], Charles M. Brooks[1] & Julia A. Mundy [1] ✉

The layered square-planar nickelates, $Nd_{n+1}Ni_nO_{2n+2}$, are an appealing system to tune the electronic properties of square-planar nickelates via dimensionality; indeed, superconductivity was recently observed in $Nd_6Ni_5O_{12}$ thin films. Here, we investigate the role of epitaxial strain in the competing requirements for the synthesis of the $n = 3$ Ruddlesden-Popper compound, $Nd_4Ni_3O_{10}$, and subsequent reduction to the square-planar phase, $Nd_4Ni_3O_8$. We synthesize our highest quality $Nd_4Ni_3O_{10}$ films under compressive strain on $LaAlO_3$ (001), while $Nd_4Ni_3O_{10}$ on $NdGaO_3$ (110) exhibits tensile strain-induced rock salt faults but retains bulk-like transport properties. A high density of extended defects forms in $Nd_4Ni_3O_{10}$ on $SrTiO_3$ (001). Films reduced on $LaAlO_3$ become insulating and form compressive strain-induced $c$-axis canting defects, while $Nd_4Ni_3O_8$ films on $NdGaO_3$ are metallic. This work provides a pathway to the synthesis of $Nd_{n+1}Ni_nO_{2n+2}$ thin films and sets limits on the ability to strain engineer these compounds via epitaxy.

The discovery of superconductivity in infinite-layer $Nd_{0.8}Sr_{0.2}NiO_2$ thin films reignited an interest in the nickelates as cuprate analogues[1–11]. More broadly, the infinite-layer nickelates are the $n = \infty$ member of a homologous series of 'layered square-planar nickelates', $R_{n+1}Ni_nO_{2n+2}$ or $(RNiO_2)_n(RO_2)$, where $R$ = trivalent rare-earth cation and $n > 1$. These compounds host $n$ quasi-two-dimensional $NiO_2$ planes separated by $(RO_2)^-$ spacer layers, as illustrated in Fig. 1. Consequently, the layering $n$ tunes the nickel $3d$ electron filling. Mapped onto the cuprate phase diagram, the bulk stable $n = 3$ compound, $Nd_4Ni_3O_8$, lies in the overdoped regime with a formal electron count of $3d^{8.67}$. Indeed, $Pr_4Ni_3O_8$ single crystals are metallic[12], as corroborated by first-principles calculations[13]. The $n = 5$ compound, $Nd_6Ni_5O_{12}$, has a formal electron count of $3d^{8.8}$ aligned with optimal doping; thin films were recently found to be superconducting[14]. The layering $n$ also tunes the out-of-plane electronic dispersion: density-functional theory (DFT) calculations suggest that, despite their similar $d$ electron fillings, the electronic structure of $Nd_6Ni_5O_{12}$ is more two-dimensional, and thus more cuprate-like, than that of the hole-doped infinite-layer nickelates[12,14,15]. Proposals to further promote cuprate-like electronic structure and enhance $T_c$ in the nickelates include electron doping the lower dimensional $n = 3$ compound[12,14], decreasing the $c$-axis lattice constant[16], increasing compressive strain via epitaxy[10,11], and applying pressure[17]. Furthermore, studies of layered square-planar nickelates in powder[18–22] and single crystal form have revealed a cuprate-like Fermi

[1]Department of Physics, Harvard University, Cambridge, MA, USA. [2]School of Applied and Engineering Physics, Cornell University, Ithaca, NY, USA. [3]Kavli Institute at Cornell for Nanoscale Science, Cornell University, Ithaca, NY, USA. [4]Department of Physics, Arizona State University, Tempe, AZ, USA. [5]The Rowland Institute, Harvard University, Cambridge, MA, USA. [6]School of Engineering and Applied Science, Harvard University, Cambridge, MA, USA. [7]Department of Chemistry and Chemical Biology, Harvard University, Cambridge, MA, USA. [8]Advanced Light Source, Lawrence Berkeley National Laboratory, Berkeley, CA, USA. [9]Platform for the Accelerated Realization, Analysis, and Discovery of Interface Materials (PARADIM), Cornell University, Ithaca, NY, USA. [10]Present address: School of Electrical and Computer Engineering, University of Oklahoma, Norman, OK, USA. [11]These authors contributed equally: Dan Ferenc Segedin, Berit H. Goodge. ✉e-mail: mundy@fas.harvard.edu

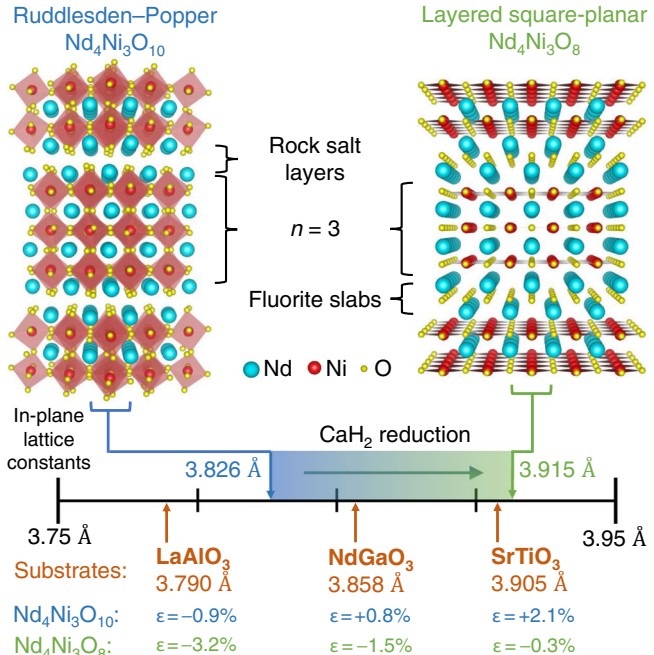

**Fig. 1 | Schematic crystal structures of (left) Nd$_4$Ni$_3$O$_{10}$ ($n$ = 3, Ruddlesden–Popper) and (right) Nd$_4$Ni$_3$O$_8$ ($n$ = 3, layered square-planar).** Neodymium, nickel, and oxygen atoms are depicted in blue, red, and yellow, respectively. The number line presents the bulk in-plane lattice parameters of Nd$_4$Ni$_3$O$_{10}$ (3.826 Å)[95] and Nd$_4$Ni$_3$O$_8$ (3.915 Å)[20], as well as the pseudocubic lattice parameters of LaAlO$_3$ (001), NdGaO$_3$ (110), SrTiO$_3$ (001). The lattice mismatch between Nd$_4$Ni$_3$O$_{10}$, Nd$_4$Ni$_3$O$_8$ and the three substrates are shown in blue and green, respectively.

surface[23], charge/spin stripes[24,25], orbital polarization[12], and large superexchange[26,27]. Layered square-planar nickelate thin films thus form an exciting platform to investigate the role of dimensionality, epitaxial strain, and chemical doping in nickelate superconductivity.

The synthesis of square-planar nickelate thin films, however, remains an immense challenge. Due to their low decomposition temperatures, infinite-layer and layered square-planar nickelates are accessible only via low temperature topotactic reduction of the perovskite $R$NiO$_3$ ($n = \infty$) or Ruddlesden–Popper $R_{n+1}$Ni$_n$O$_{3n+1}$ ($n > 1$) parent compounds, respectively[18,18,19,28–31]. Furthermore, the absence of superconductivity in reduced nickelate powders[32–34] and bulk single crystals[35] to date suggests that external stabilization by a substrate may be required to yield superconductivity. The synthesis of reduced nickelate thin films, however, is complicated by the large ~2–3% increase in the in-plane lattice parameter upon reduction. To minimize compressive strain in the reduced phase, the parent compound must be synthesized under tensile strain, as shown in Fig. 1. Perovskite nickelates synthesized under ~2.6% tensile strain on SrTiO$_3$, however, exhibit strain-relieving extended defects[3,6,9,36,37]. A dramatic reduction of such extended defects has recently been achieved by synthesizing the parent perovskite under more modest ~1.6% tensile strain on (LaAlO$_3$)$_{0.3}$(Sr$_2$TaAlO$_6$)$_{0.7}$ (LSAT)[10,11]. Extended defects can also form through the reduction process. For example, LaNiO$_2$/SrTiO$_3$ exhibits $a$-axis oriented domains which relieve the 1.4% compressive strain[6], while these domains are absent in NdNiO$_2$/SrTiO$_3$ with 0.4% compressive strain[37]. Therefore, understanding the strain-dependent stability of both the parent Ruddlesden–Popper and reduced square-planar phases is essential to optimize the synthesis of layered square-planar nickelate thin films.

Here, we discuss the competing requirements for the synthesis and oxygen deintercalation of Nd$_4$Ni$_3$O$_{10}$ thin films on LaAlO$_3$ (001), NdGaO$_3$ (110), and SrTiO$_3$ (001). We focus on the $n$ = 3

Ruddlesden–Popper compound because both the oxidized Nd$_4$Ni$_3$O$_{10}$ and reduced Nd$_4$Ni$_3$O$_8$ compounds have been synthesized as bulk single crystals, allowing us to benchmark the strain states with bulk lattice constants. Figure 1 tabulates the lattice mismatch, $\epsilon = (a_{sub} - a_{bulk})/a_{bulk}$, of Nd$_4$Ni$_3$O$_{10}$ and Nd$_4$Ni$_3$O$_8$ on LaAlO$_3$, NdGaO$_3$, and SrTiO$_3$. We present the molecular beam epitaxy (MBE) synthesis of Nd$_4$Ni$_3$O$_{10}$ on the three substrates. We show that Nd$_4$Ni$_3$O$_{10}$ on SrTiO$_3$ exhibits a high degree of disorder characterized by a near-equal density of vertical and horizontal rock salt faults; consequently, we do not consider these films for reduction. When synthesized under lesser tensile strain on NdGaO$_3$, a smaller density of vertical rock salt faults forms while maintaining relatively high-quality Ruddlesden–Popper ordering. By contrast, Nd$_4$Ni$_3$O$_{10}$ on LaAlO$_3$ exhibits coherent ordering of horizontal rock salt layers with very few extended defects.

Next, we reduce Nd$_4$Ni$_3$O$_{10}$ to the square-planar phase, Nd$_4$Ni$_3$O$_8$, on LaAlO$_3$ and NdGaO$_3$. In Nd$_4$Ni$_3$O$_8$ on LaAlO$_3$, we observe regions with pristine square-planar ordering along with disordered regions where the $c$-axis cants locally by as much as 7°, likely a compressive strain relaxation mechanism[6,38]. However, all reduced films on LaAlO$_3$ are insulating. On the other hand, Nd$_4$Ni$_3$O$_8$ and Nd$_6$Ni$_5$O$_{12}$ on NdGaO$_3$ are metallic and superconducting[14], respectively. Furthermore, our density-functional theory (DFT) calculations reveal that in-plane strain alters aspects of the $R_{n+1}$Ni$_n$O$_{2n+2}$ electronic structure relevant to superconductivity. Our study thus demonstrates a pathway to the synthesis of a superconducting compound and sets limits on the ability to strain-engineer $R_{n+1}$Ni$_n$O$_{2n+2}$ thin films via epitaxy.

## Results

### Density-functional theory calculations

We perform DFT calculations to investigate how strain tunes the electronic structure of Nd$_4$Ni$_3$O$_8$ and Nd$_6$Ni$_5$O$_{12}$. As detailed in Supplementary Note 1, our calculations reveal the following effects:

- The charge-transfer energy ($\Delta$, a measure of the $p$–$d$ splitting) increases (decreases) with compressive (tensile) strain.
- The rare-earth density of states at the Fermi level increases (decreases) with compressive (tensile) strain.
- The Ni-$d_{x^2-y^2}$ bandwidth increases (decreases) with compressive (tensile) strain.

The role of the charge-transfer energy $\Delta$ and rare-earth 5$d$ 'spectator' bands in nickelate superconductivity has been extensively debated[39–42]. In the cuprates, the superexchange interaction $J \sim t^4/\Delta^3$ and single-band fermiology are widely deemed essential for superconductivity[43,44]. Thus strain-engineering $R_{n+1}$Ni$_n$O$_{2n+2}$ compounds provide an avenue to explore the role of the charge-transfer energy and multi- or single-band fermiology in nickelate and cuprate superconductivity. Next, we discuss the synthesis of these compounds under a variety of strain states.

### MBE synthesis of Ruddlesden–Popper nickelates

The synthesis of Ruddlesden–Popper nickelate thin films shares several of the challenges encountered in the synthesis of perovskite nickelates, including the difficulty in reaching the high oxidation Ni$^{3+}$ state and the formation of secondary phases like NiO[37,45–47]. There are, however, additional challenges unique to the synthesis of Ruddlesden–Popper thin films. In general, the MBE synthesis of $A_{n+1}B_nO_{3n+1}$ Ruddlesden–Popper compounds requires the precise sequential deposition of $A$ and $B$ monolayers to achieve the desired composition ($A/B = (n+1)/n$) and monolayer dose[47–49]. Given ideal composition, errors in the monolayer dose times result in the formation of Ruddlesden–Popper layers with an average periodicity different from the one targeted[49]. Errors in composition, on the other hand, can be accommodated via the formation of extended defects such as rock salt faults: half unit cell stacking faults formed by additional $A$O inclusions[50–52]. In homoepitaxial Sr$_{n+1}$Ti$_n$O$_{3n+1}$, for example, up to 5%

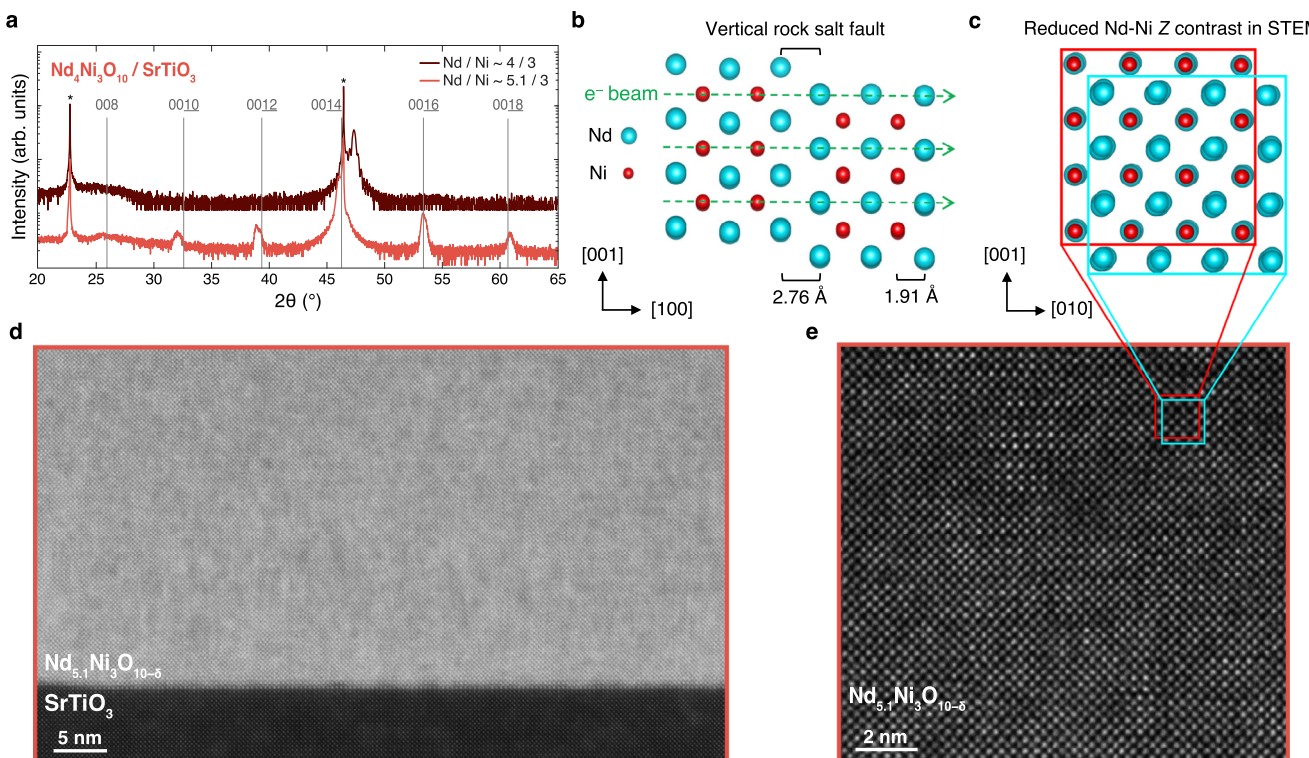

**Fig. 2 | Structural characterization of Nd4Ni3O10/SrTiO3 (001) ($\varepsilon$ = + 2.1%). a** XRD scans of Nd4Ni3O10/SrTiO3 with Nd/Ni ~ 4/3 (nominally stoichiometric) and Nd/Ni ~ 5.1/3 (27.5% neodymium-rich). These compositions are measured by Rutherford backscattering spectroscopy (RBS). The asterisks denote substrate peaks and the vertical lines mark the 00$l$ peak positions of bulk Nd4Ni3O10[95]. **b** Schematic crystal structure depicting a vertical rock salt fault. **c** 90° rotation of the crystal structure in (b) illustrating the origin of the atomic contrast loss in STEM. The green dashed arrows in (**b**) and circled dots in (**c**) denote the propagation direction of the electron beam during STEM measurements. **d** MAADF-STEM image of the film with Nd/Ni ~ 5.1/3 in (**a**). **e** Atomic-resolution HAADF-STEM image showing the representative lattice structure of the film. The reduced atomic contrast is due to projection through vertically offset Ruddlesden–Popper regions, shown schematically in (**c**). We label the oxygen stoichiometry as 'O10-$\delta$' because the precise oxygen content of this film is unknown.

excess strontium can be accommodated by the formation of additional SrO rock salt layers, while strontium deficiency results in missing rock salt layers[53–57].

In addition to the composition and monolayer dose, lattice mismatch plays a crucial role in the formation of extended defects. In compressively strained LaNiO3/LaAlO3 films, for example, misfit dislocations form to relax the in-plane compressive strain[58,59]. Synthesized under tensile strain on SrTiO3, LaNiO3[58] and NdNiO3[36] films exhibit vertical rock salt faults. These extended defects effectively increase the in-plane lattice constant and thus relieve tensile strain because the distance between rare-earth planes within a rock salt layer (~2.8 Å) is greater than the atomic layer spacing of perovskite (La, Nd)NiO3 (~1.9 Å) (see Supplementary Note 2). The density of vertical rock salt faults in infinite-layer nickelates has been decreased dramatically by decreasing the tensile strain of the parent perovskite compound[10,11]. In comparison to perovskites, however, Ruddlesden–Popper compounds are even more prone to the formation of vertical rock salt faults because their structure already hosts horizontal rock salt layers, which can instead orient vertically to relieve the in-plane tensile strain.

Most infinite-layer nickelate films to date have been synthesized on SrTiO3, motivated by the minimal 0.4% compressive strain in the reduced state. Here, we present attempts to synthesize Nd4Ni3O10 under 2.1% tensile strain on SrTiO3. In Fig. 2a, we present x-ray diffraction (XRD) scans of two Nd4Ni3O10/SrTiO3 films: one with Nd/Ni ~ 4/3 and the other with Nd/Ni ~ 5.1/3, a 27.5% excess in neodymium content. We will refer to these two films as 'stoichiometric' and 'neodymium-rich', respectively. The stoichiometric film exhibits two primary features in the XRD scan: a film peak at 47.4° and a broad peak at ~26°. These features are reminiscent of disordered and non-stoichiometric

$R$NiO3[7,60,61]. However, the stoichiometric film exhibits none of the 00$l$ superlattice peaks expected for bulk Nd4Ni3O10. The neodymium-rich film, on the other hand, exhibits superlattice peaks that are roughly consistent with bulk Nd4Ni3O10. Thus, coherent horizontal rock salt ordering forms only after supplying excess neodymium.

We investigate the nature of this crystalline disorder by atomic-resolution scanning transmission electron microscopy (STEM) imaging of the film. Microscopic signatures of Ruddlesden–Popper disorder evident by STEM include a zigzag arrangement of $A$-cations and reduced $A$-$B$ contrast[36,62,63]. As illustrated in Fig. 2b, a vertical rock salt fault in the (001) plane leads to the relative shift of an adjacent rock salt layer by $a$/2 [011][64]. Consequently, the neodymium and nickel atomic sites are stacked in projection along the propagation direction of the electron beam, resulting in reduced atomic number ($Z$) contrast[57] (Fig. 2c). We present a medium-angle annular dark field (MAADF)-STEM image of the Nd5.1Ni3O10-$\delta$ film in Fig. 2d. The film is distinguished from the substrate by the bright contrast which arises from the heavy neodymium nuclei in the film. Within the film, however, the clean layered perovskite structure of an $n$ = 3 Ruddlesden–Popper phase is obscured by a high density of disordered vertical and horizontal rock salt planes. A higher magnification (HAADF)-STEM image of the film shown in Fig. 2e similarly indicates a high density of rock salt plane formation through the reduced $Z$ contrast. A near-equal density of vertical and horizontal rock salt faults thus forms in spite of the sequential MBE shuttering sequence encouraging horizontal rock salt layering.

Our attempts to synthesize Nd4Ni3O10 on SrTiO3 demonstrate the extraordinary difficulties in stabilizing coherent horizontal rock salt ordering under large tensile strain. High-quality perovskite NdNiO3, on

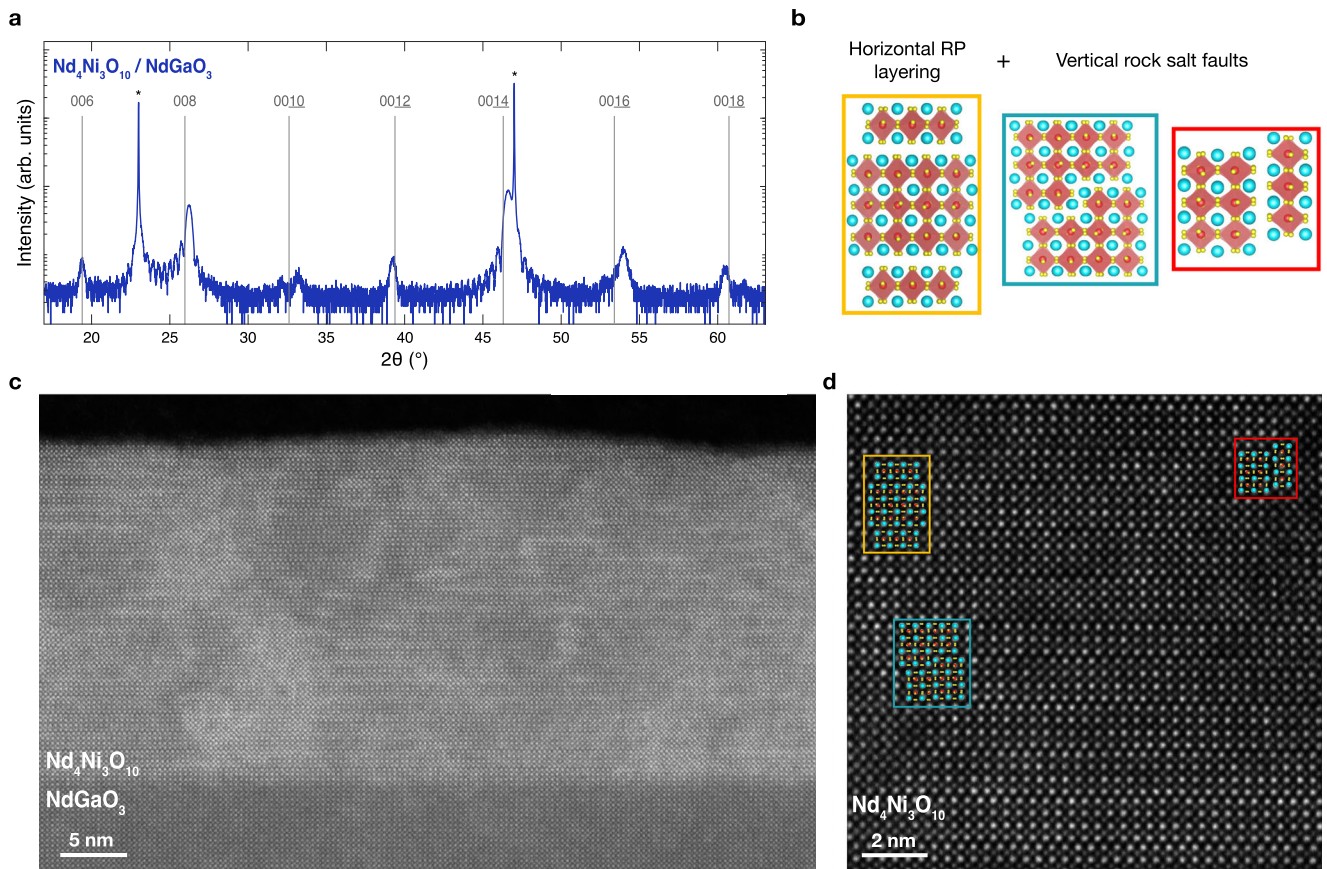

**Fig. 3 | Structural characterization of Nd$_4$Ni$_3$O$_{10}$/NdGaO$_3$ (110) ($\epsilon$ = + 0.8%).**
**a** XRD scan of a Nd$_4$Ni$_3$O$_{10}$/NdGaO$_3$ film. The asterisks denote substrate peaks and the vertical lines mark the 00$l$ peak positions of bulk Nd$_4$Ni$_3$O$_{10}$[95]. **b** Schematic crystal structures depicting three regions in (**d**). **c** MAADF-STEM image of the Nd$_4$Ni$_3$O$_{10}$/NdGaO$_3$ film shown in (**a**). **d** Atomic-resolution HAADF-STEM image showing the representative lattice structure of the film with atomic model overlays. The yellow box highlights horizontal rock salt ordering while the green and red boxes show regions with half and three unit cell long vertical rock salt faults, respectively.

the other hand, can be synthesized on SrTiO$_3$, albeit with a small density of vertical rock salt faults[36,37,58]. Ruddlesden−Popper nickelates, however, are more likely to form vertical rock salt faults than perovskites due to the composition−in NdNiO$_3$, Nd/Ni = 1/1, while in Nd$_4$Ni$_3$O$_{10}$, Nd/Ni = 4/3. We thus propose that the additional neodymium content in Nd$_4$Ni$_3$O$_{10}$ under tensile strain forms strain-relieving vertical rock salt faults instead of horizontal rock salt layers, regardless of the MBE shuttering sequence. In fact, to stabilize any horizontal rock salt ordering, it is necessary to supply an excess of neodymium, as demonstrated in Fig. 2a. Ruddlesden−Popper films are therefore more sensitive to the formation of tensile strain-relieving extended defects than their perovskite counterparts. Additionally, SrTiO$_3$ is unique among the substrates included in our study for its charge neutral atomic planes which form a stronger discontinuity to the charged planes in the Nd$_4$Ni$_3$O$_{10}$ film. Spectroscopic and theoretical studies of the interface between NdNiO$_2$ and SrTiO$_3$ show that a similar polar discontinuity is alleviated by a unit cell thick reconstruction layer of Nd(Ti,Ni)O$_3$[65], which we also observe in our Nd$_4$Ni$_3$O$_{10}$ films on SrTiO$_3$ (Supplementary Fig. S23). Future studies may reveal subtle differences in polar accommodation of infinite- and several-layer nickelates. In this work we focus on epitaxial strain as the primary driver of reduced crystalline quality in these films. Given the high density of extended defects observed in Nd$_4$Ni$_3$O$_{10}$/SrTiO$_3$, we disqualify this system from consideration for reduction and instead turn our attention to substrates which yield smaller lattice mismatch.

In an effort to decrease the density of extended defects, we next synthesize Nd$_4$Ni$_3$O$_{10}$ under more modest 0.8% tensile strain on NdGaO$_3$ (Fig. 1). We present an XRD scan of a Nd$_4$Ni$_3$O$_{10}$/NdGaO$_3$ film in Fig. 3a which exhibits 00$l$ superlattice peaks consistent with bulk Nd$_4$Ni$_3$O$_{10}$. The splitting of the 00<u>10</u> peak may be indicative of a small error in the monolayer dose or composition[49]. Reciprocal space mapping in Supplementary Fig. S22 demonstrates that the film is epitaxially strained to the substrate. Cross-sectional STEM imaging in Fig. 3c, d provides a microscopic view of the defects. Diffraction contrast near the rock salt planes in the large field-of-view MAADF-STEM image in Fig. 3c highlights coherent horizontal ordering of Ruddlesden−Popper layers and additional vertical rock salt faults. A HAADF-STEM image of the film in Fig. 3d similarly identifies the coexistence of different Ruddlesden−Popper phases as well as vertical rock salt faults. Boxes and atomic models in Fig. 3b, d denote regions of well-ordered horizontal Ruddlesden−Popper layers (yellow), vertical rock salt faults (red), and local step edges between regions of mixed local $n$ phase (green). Therefore, while some vertical rock salt faults are observed, the lower tensile strain imparted by NdGaO$_3$ reduces the density of vertical Ruddlesden−Popper faults and other extended defects that dominate Nd$_4$Ni$_3$O$_{10}$ films synthesized on SrTiO$_3$.

To determine whether vertical rock salt fault formation can be further mitigated, we next synthesize Nd$_4$Ni$_3$O$_{10}$ under 0.9% compressive strain on LaAlO$_3$. The XRD scan of a Nd$_4$Ni$_3$O$_{10}$/LaAlO$_3$ film in Fig. 4a exhibits sharp superlattice peaks with no peak splitting. Furthermore, reciprocal space mapping in Supplementary Fig. S12 confirms the film is epitaxially strained to the substrate. More detailed structural and electronic characterization of our Ruddlesden−Popper

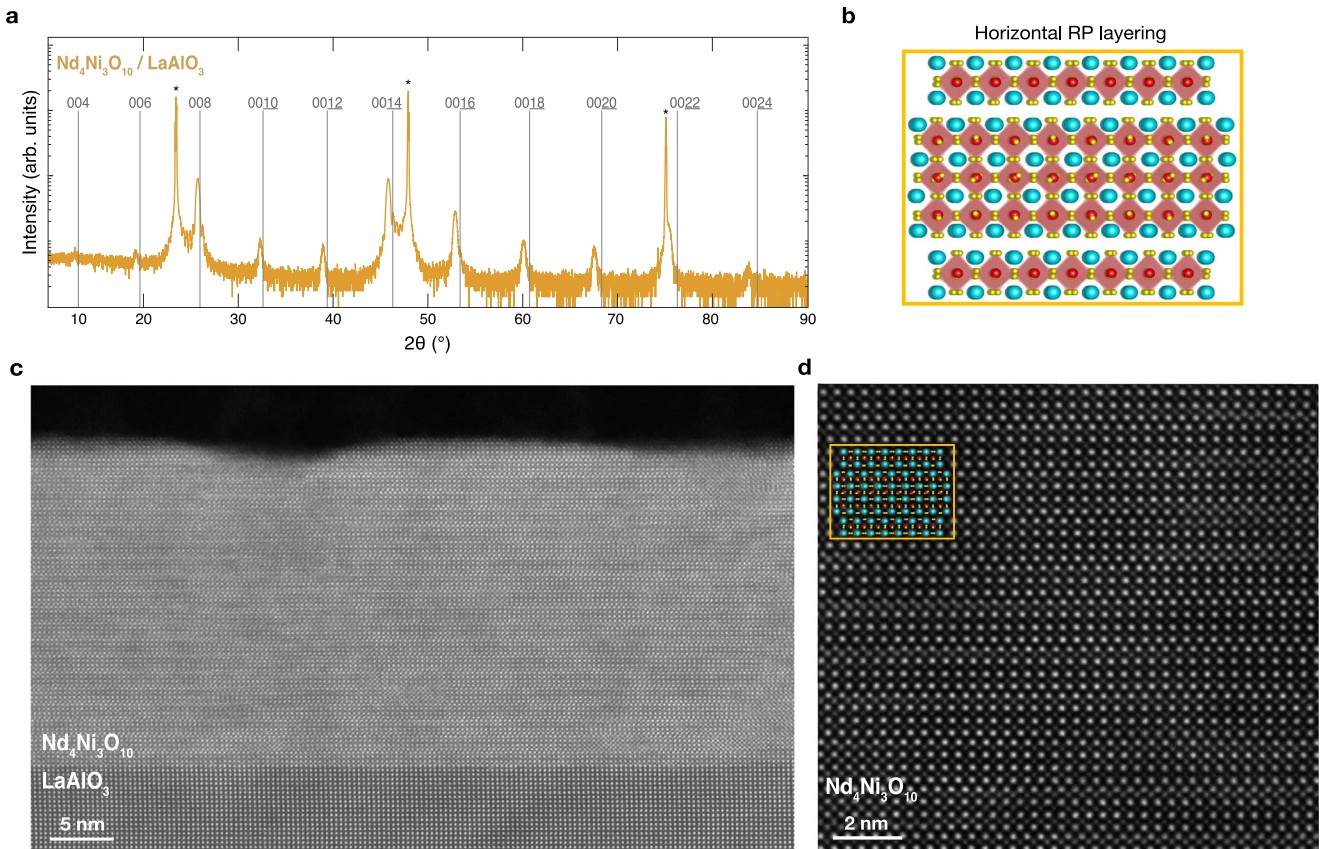

**Fig. 4 | Structural characterization of $Nd_4Ni_3O_{10}/LaAlO_3$ (001) ($\epsilon = -0.9\%$).**
**a** XRD scan of a $Nd_4Ni_3O_{10}/LaAlO_3$ film. The asterisks denote substrate peaks and the vertical lines mark the $00l$ peak positions of bulk $Nd_4Ni_3O_{10}$[95]. **b** Schematic crystal structure of well-ordered, horizontal Ruddlesden–Popper layers. **c** MAADF-STEM image of the $Nd_4Ni_3O_{10}/LaAlO_3$ film shown in (**a**). **d** Atomic-resolution HAADF-STEM image showing the representative lattice structure of the film with an atomic model overlay of the structure shown in (**b**), boxed in yellow.

($n = 1 - 5$) nickelates on $LaAlO_3$ can be found in ref. [47]. MAADF-STEM images of this film in Fig. 4c show nearly uniform adherence to horizontal layering with very few vertical rock salt faults. Atomic-resolution HAADF-STEM imaging in Fig. 4d corroborates the high degree of crystalline order with only a few defects visible. Deviations from the $n = 3$ Ruddlesden–Popper structure are quantified in Supplementary Fig. S7 in ref. [47]. While such deviations from the targeted $n = 3$ Ruddlesden–Popper structure are observed in $La_4Ni_3O_{10}$[66,67] and reduced $Nd_4Ni_3O_8$[68] bulk crystals, our films exhibit a small density of such deviations due to the MBE shuttering sequence that encourages the formation of the targeted Ruddlesden–Popper order.

We present a summary of the characteristic crystalline microstructure for $Nd_4Ni_3O_{10}$ films under varying amounts of compressive and tensile epitaxial strain in Fig. 5. Strain analysis of the (101) and ($\bar{1}$01) pseudocubic lattice fringes highlight local $a/2$ (011) lattice offsets at both horizontal and vertical Ruddlesden–Popper rock salt planes. Under 0.9% compressive strain on $LaAlO_3$, $Nd_4Ni_3O_{10}$ exhibits coherent horizontal Ruddlesden–Popper ordering with some rock salt discontinuities but very few vertical rock salt faults (Fig. 5a). Under 0.8% tensile strain on $NdGaO_3$, the horizontal Ruddlesden–Popper layering structure is largely preserved with the emergence of some vertical rock salt planes (Fig. 5b). The density of vertical rock salt faults in $Nd_4Ni_3O_{10}$ increases dramatically as the tensile strain is increased to $\epsilon = +2.1\%$ on $SrTiO_3$, with a near-equal density of vertical and horizontal rock salt faults evident in Fig. 5c. Such an increase in rock salt fault density is expected as the vertical Ruddlesden–Popper layers relieve tensile strain[36,58] (see Supplementary Note 2). With these challenges in stabilizing the parent Ruddlesden–Popper compounds in mind, we next consider their oxygen deintercalation. Crucially, much of the cation

disorder observed in the parent compounds is preserved through reduction due to the minimal cation mobility at typical reduction temperatures (~300 °C).

## Oxygen deintercalation to the layered square-planar phase

A fundamental issue in the topotactic reduction of nickelates is the metastability of square-planar phases at typical reduction temperatures (~300 °C)[18–20,30,69,70]. Decomposition to $R_2O_3$ ($R$ = La, Pr, Nd), NiO, and nickel metal has been reported at temperatures as low as 210 °C for $LaNiO_2$[30] and 200° C for $NdNiO_2$[31]. For layered square-planar compounds, the decomposition temperatures are higher: 400 °C for $La_4Ni_3O_8$[18] and 375 °C for $La_3Ni_2O_7$[19]. This difference may be ascribed to the suppressed out-of-plane cation mobility in Ruddlesden–Popper and layered square-planar nickelates, facilitated by the rock salt $R$O or fluorite $R$O$_2$ 'blocking' layers, respectively. While nickelate powders can be reduced at temperatures as low as 190 °C with metal hydrides[30], nickelate thin films typically require higher reaction temperatures (>250 °C) to stabilize the square-planar phase[37,38], possibly because films, unlike powders, cannot be ground with the reductant. As a result, decomposition has been observed in reduced infinite-layer thin films[37,71–73]; the precise identification of the decomposition products is, however, difficult in thin films in part due to the small film volume. An additional challenge associated with metal hydride reductions is the possibility of hydrogen incorporation in the lattice, an effect which has been proposed as a reason for the absence of superconductivity in some films[70,74,75]. For example, an oxyhydride $NdNiO_xH_y$ phase was reported in $NdNiO_3$ films reduced with $CaH_2$[71], but to date no experiment has linked insulating behavior with hydrogen intercalation. Solid-state reduction techniques have recently been demonstrated as a

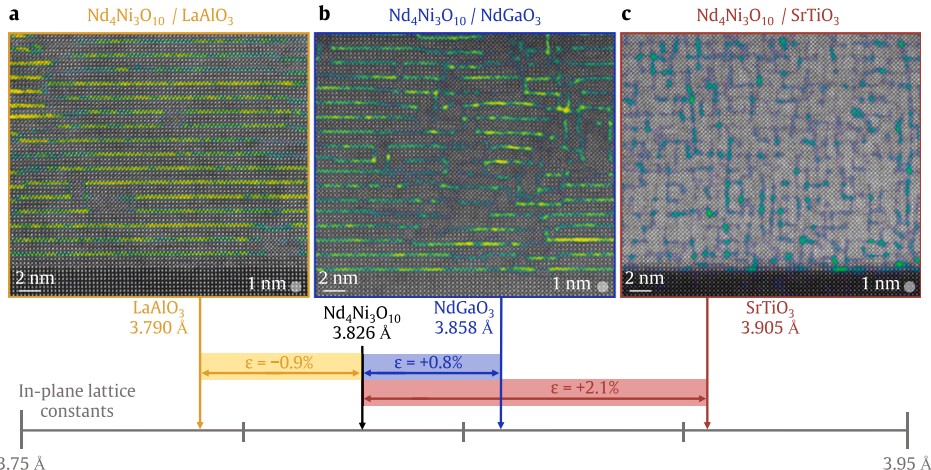

**Fig. 5 | Strain-dependent structural characterization of Nd$_4$Ni$_3$O$_{10}$ films.** Lattice fringe strain maps highlighting the characteristic formation of horizontal and vertical rock salt layers in (**a**) Nd$_4$Ni$_3$O$_{10}$/LaAlO$_3$ (001), (**b**) Nd$_4$Ni$_3$O$_{10}$/NdGaO$_3$ (110), and (**c**) Nd$_4$Ni$_3$O$_{10}$/SrTiO$_3$ (001) films. The number line depicts the in-plane lattice constant of Nd$_4$Ni$_3$O$_{10}$ in black and the corresponding strain states on LaAlO$_3$ in yellow, NdGaO$_3$ in blue, and SrTiO$_3$ in red. More details of the analysis are provided in Methods. Raw STEM data and strain map outputs are shown in Supplementary Fig. S26. See Supplementary Fig. S4 for a discussion regarding the observed strain-dependent defect formation.

promising alternative to metal hydride reductions[76]. Thus to minimize the formation of decomposition products and potential hydrogen intercalation during metal hydride reductions, careful optimization of reduction duration and temperature is essential.

Additionally, the dramatic expansion (contraction) of the in-plane (out-of-plane) lattice parameter through reduction further complicates the stabilization of high-quality square-planar nickelates. After reduction, La$_{1-x}$Ca$_x$NiO$_2$ single crystals exhibit three orthogonally oriented domains of the infinite-layer phase separated by microcracks[35]. Nevertheless, La$_{1-x}$Ca$_x$NiO$_2$ and Pr$_4$Ni$_3$O$_8$[35] single crystals are metallic but not yet superconducting to date. In contrast, most reduced infinite-layer[32–34,77] and layered nickelate powders[19–22,78–80] are insulating—metallic (Pr, Nd)$_4$Ni$_3$O$_8$ pellets were, however, achieved with reductions using a sulfur getter to promote the removal of apical oxygens[81,82]. Thus metallicity in square-planar, or $T'$, nickelates may be exquisitely sensitive to the oxygen content like in $T'$ cuprates[83]. Additional proposals for the the lack of superconductivity in bulk-reduced nickelates include nickel deficiency and the nucleation of ferromagnetic nickel grains[84].

Thin films, on the other hand, have several advantages that address these issues. First, thin films host an inherent in-plane versus out-of-plane anisotropy defined by the substrate that encourages the formation of a single orientation of the reduced phase. Furthermore, the thin film geometry minimizes the size of potential ferromagnetic nickel clusters and promotes reduction uniformity due to the large surface area exposed to the reductant. Thin films, however, face a challenge absent in bulk compounds: defects may form in response to reduction-induced compressive strain. In LaNiO$_2$/SrTiO$_3$, for example, macroscopic[38] and local[6,73] $a$-axis oriented infinite-layer domains form, likely to relieve the 1.4% compressive strain[6]. This effect was also demonstrated in LaNiO$_2$ films which exhibit an increasing fraction of $a$-axis oriented LaNiO$_2$ with increasing compressive strain[72]. While strain is an important factor in defect formation, synthesis optimization of the parent compound has suppressed the formation of such defects upon reduction[6,11]. The reduction-induced strain-relieving mechanisms in layered nickelates may be different than those observed in infinite-layer nickelates.

With these challenges in mind, we now turn to reductions on LaAlO$_3$, which yield the highest quality as-synthesized Ruddlesden–Popper nickelates (Fig. 5), but also the highest compressive strain in the reduced state (Fig. 1). We first reduce perovskite NdNiO$_3$ on LaAlO$_3$ to study how a system without horizontal rock salt layers undergoes an increase in compressive strain from 0.5% to 3.3% upon reduction. We present the XRD scans of an as-synthesized NdNiO$_3$ and reduced NdNiO$_2$ film on LaAlO$_3$ in Fig. 6a. The as-synthesized film exhibits the 002 NdNiO$_3$ peak at ~47.6° with no second phases evident over the full scan range (Supplementary Fig. S10). In the reduced film, we observe a peak at ~46.3° along with a lower intensity peak at ~56.0°, which likely correspond to the 200 and 002 peaks of NdNiO$_2$, respectively. Incremental reduction of the film results in the immediate formation of the $a$-axis oriented NdNiO$_2$ phase (Supplementary Fig. S10), which suggests that this phase is not formed a result of over-reduction, as was observed in LaNiO$_2$/SrTiO$_3$[38]. Furthermore, NdNiO$_2$/LaAlO$_3$ exhibits insulating electrical transport (Supplementary Fig. S10). These results suggest that NdNiO$_3$ undergoes large-scale $a$-axis reorientation upon reduction under high compressive strain, as illustrated in Fig. 6b.

Next, we reduce Nd$_4$Ni$_3$O$_{10}$ on LaAlO$_3$ to investigate how rock salt layering impacts the structural transformation to the square-planar phase. The reduction-induced increase in the compressive strain of the Ruddlesden–Popper (from 0.9% to 3.2%) is comparable to that of the perovskite (from 0.5% to 3.3%). In Fig. 6c, we present XRD scans of Nd$_4$Ni$_3$O$_{10}$ and Nd$_4$Ni$_3$O$_8$ films on LaAlO$_3$. The parent film exhibits all expected Nd$_4$Ni$_3$O$_{10}$ 00$l$ peaks and, as shown in Fig. 6e, a metal-to-insulator transition at ~150 K consistent with previous reports[46,47]. Upon reduction, the Nd$_4$Ni$_3$O$_{10}$ 00$l$ peaks shift toward the Nd$_4$Ni$_3$O$_8$ 00$l'$ peaks, revealing the formation of the square-planar phase. Furthermore, the lack of 200 NdNiO$_2$ or 220 Nd$_4$Ni$_3$O$_8$ peaks at ~46.3° suggests that the film has retained $c$-axis orientation through reduction, as illustrated in Fig. 6d. Reciprocal space maps in Supplementary Fig. S12 however demonstrate that the Nd$_4$Ni$_3$O$_8$ film is partially relaxed with a 3.89 Å in-plane lattice constant compared to the 3.915 Å bulk value. XAS at the oxygen K-edge in Supplementary Fig. S13 further corroborates the complete reduction to the square-planar phase. Thus, in contrast to NdNiO$_3$, Nd$_4$Ni$_3$O$_{10}$ retains global $c$-axis orientation upon reduction, despite the high compressive strain on LaAlO$_3$. Like NdNiO$_2$/LaAlO$_3$, however, the reduced layered nickelate is insulating (Fig. 6e). In fact, all films reduced on LaAlO$_3$ are insulating, likely due to reduction-induced structural disorder (Supplementary Note 4). Furthermore, of the three strain states studied here, $R_{n+1}$Ni$_n$O$_{2n+2}$ films under high compressive strain on LaAlO$_3$ are the furthest from cuprate-like due to the

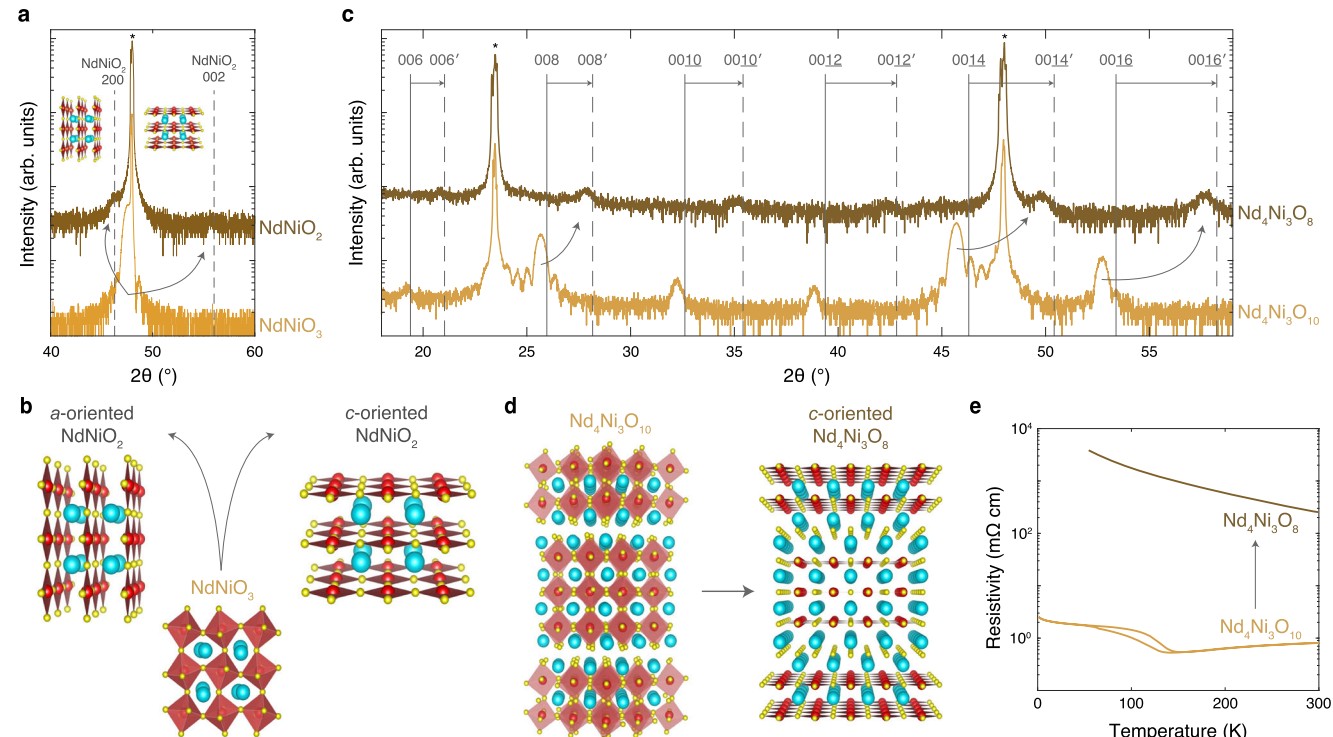

**Fig. 6 | Structural and electrical transport characterization of NdNiO₃ and Nd₄Ni₃O₁₀ reductions on LaAlO₃ (001). a** XRD scans of a 15.5 nm NdNiO₃ film as-synthesized (bottom) and reduced for 3 hours at 290 °C (top). The vertical dashed lines denote the 200 and 002 peak positions of bulk NdNiO₂[31]. **b** Schematic crystal structures illustrating the formation of a mixture of *a*- and *c*-axis oriented NdNiO₂ upon reduction of NdNiO₃. **c** XRD scans of a 20.0 nm Nd₄Ni₃O₁₀ film as-synthesized (bottom) and Nd₄Ni₃O₈ reduced for 3 hours at 290 °C (top). The vertical solid and dashed lines denote 00*l* peak positions of bulk Nd₄Ni₃O₁₀[95] and Nd₄Ni₃O₈[20], respectively. The primed indices distinguish the reduced layered square-planar phase from the as-synthesized Ruddlesden–Popper[20]. **d** Schematic crystal structures illustrating the reduction of *c*-axis oriented Nd₄Ni₃O₁₀ to *c*-axis oriented Nd₄Ni₃O₈. **e** Resistivity measurements of the Nd₄Ni₃O₁₀ and Nd₄Ni₃O₈ films in (**c**).

larger charge-transfer energy and higher neodymium DOS at $\varepsilon_F$ (Supplementary Note 1). Thus if optimized to a metallic state, $R_{n+1}Ni_nO_{2n+2}$/LaAlO₃ films could be a platform to study the role of the charge-transfer energy and neodymium 5*d* states in nickelate superconductivity.

Cross-sectional STEM measurements shown in Fig. 7 reveal the microstructure of the reduced Nd₄Ni₃O₁₀/LaAlO₃ film. The large field-of-view MAADF-STEM image in Fig. 7a shows some decomposition near the surface and diagonal defects between crystalline regions throughout the film. Atomic-resolution HAADF-STEM imaging in Fig. 7(b) shows one such clean region with horizontal layer ordering similar to the as-synthesized film (Figs. 4, 5a). The reduced square-planar structure is visible by close inspection of the atomic lattice in Fig. 7c, which shows a more pronounced in- versus out-of-plane anisotropy of the pseudo-infinite-layer unit cell. Annular bright field (ABF)-STEM imaging in Fig. 7d further illustrates the reduced square-planar phase with the absence of oxygen atomic columns in the horizontal neodymium planes. The precise oxygen content of our reduced films has not been measured.

Interspersed between these regions of pristine Nd₄Ni₃O₈, we find that the diagonal defects observed by MAADF-STEM in Fig. 7a are in fact regions of local *c*-axis canting. Using local wavefitting analysis[85], in Fig. 7d we map the local *c*-axis orientation across one such diagonal defect, revealing regions of opposite canting of up to several degrees. Within this window of variation (~±10°), however, the film's *c*-axis remains globally out-of-plane. By contrast, XRD scans in Fig. 6a suggest that a perovskite NdNiO₃ film on LaAlO₃ can also be reduced to NdNiO₂, but that most of the film attains *a*-axis orientation with only a tiny peak corresponding to *c*-axis orientation remaining visible. The Ruddlesden–Popper structure therefore appears to stabilize the *c*-axis orientation even under high 3.2% compressive strain, preventing the large-scale reorientation observed in the infinite-layer counterpart on LaAlO₃.

Under more modest (1.4%) compressive strain, LaNiO₂/SrTiO₃ exhibits a small fraction of *a*-axis reorientation along diagonal planes[6], similar to those observed in the triple-layer films here. An important distinction between the infinite- and triple-layer systems is that of the heavy cation lattice and the oxygen-filled (or reduced) planes. In the perovskite/infinite-layer system, the orientation of empty oxygen planes effectively defines the local *c*-axis (orthogonal to the rare-earth planes). Without the imposition of additional symmetry by oxygen occupancy, however, the cation lattice directions are essentially equivalent within the pseudocubic approximation. Local reorientation can therefore be equivalently described as local rearrangement of occupied oxygen sites. In the Ruddlesden–Popper structure, on the other hand, the heavy cation lattice bears inherent symmetry distinctions even without the oxygen lattice, with the cation *c*-axis defined as orthogonal to the rock salt planes. Here, we observe a global preservation of this *c*-axis direction upon reduction: all the Ruddlesden–Popper layers remain in the same uniform plane from the as-synthesized to reduced phase due to minimal cation mobility at typical reduction temperatures (~300 °C). How closely, then, is the oxygen lattice tied to this pre-defined cation symmetry?

Locally, we find nanometer-scale regions near the canting defects which suggest an oxygen lattice reorientation internal to the global Ruddlesden–Popper structure. Supplementary Fig. S29 shows a high-magnification image of the defect structure mapped in Fig. 7e. The atomic overlay in the bottom right of the image—where the film is epitaxial, uncanted, and well-ordered—shows how the reduced structure can be mapped onto a rectangular sublattice with longer (shorter) in-plane (out-of-plane) atomic spacings. The rectangles outline 3 × 3 neodymium atomic sites, with the long and short dimensions colored

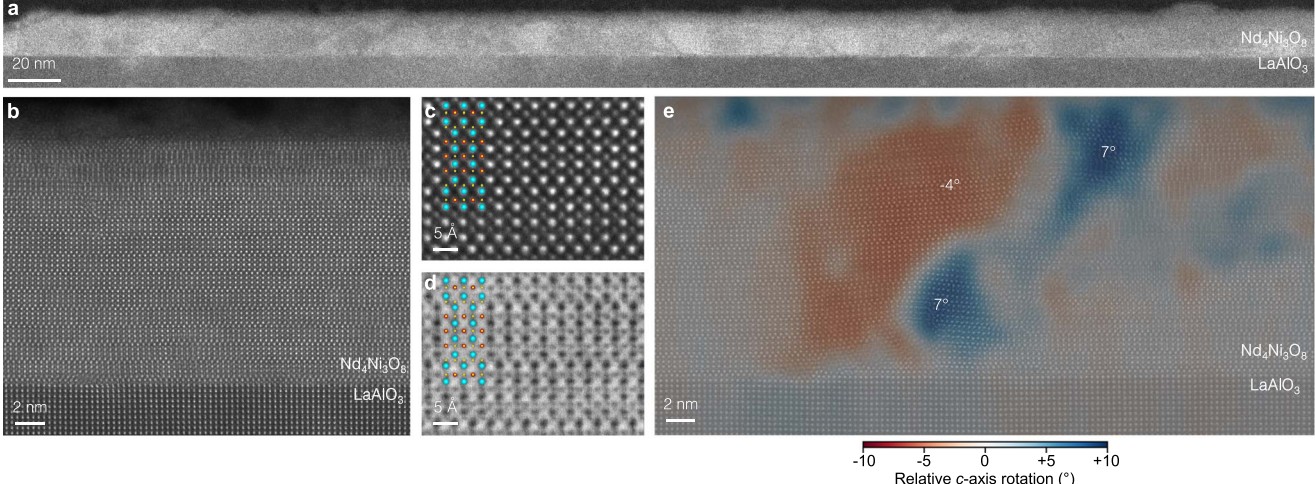

**Fig. 7 | Structural characterization of a reduced Nd₄Ni₃O₈/LaAlO₃ film. a** Large field-of-view MAADF-STEM image. **b** Atomic-resolution HAADF-STEM image of a region exhibiting high-quality layered square-planar structure. **c** HAADF and (**d**) ABF-STEM images of a small field of view with an overlaid atomic model of the square-planar structure. The full field-of-view is provided in Supplementary

Fig. S28. **e** Gaussian-smoothed map of local $c$-axis canting relative to the out-of-plane direction. The STEM image and raw tilt map are provided in Supplementary Fig. S27. The XRD and transport measurements of this film can be found in Supplementary Fig. S7.

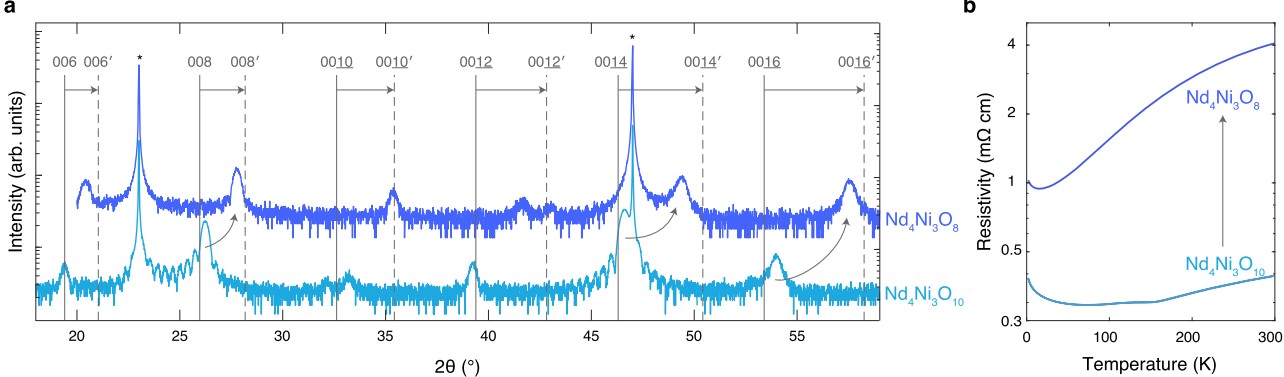

**Fig. 8 | Structural and electrical transport characterization of Nd₄Ni₃O₁₀ and Nd₄Ni₃O₈ on NdGaO₃ (110). a** XRD scans of an as-synthesized Nd₄Ni₃O₁₀ and reduced Nd₄Ni₃O₈ film on NdGaO₃. The film is 25.5 nm and reduced for 3 hours at 300 °C. The vertical solid and dashed lines denote the 00$l$ peak positions of bulk

Nd₄Ni₃O₁₀ and Nd₄Ni₃O₈, respectively. The primed indices distinguish the reduced square-planar phase from the as-synthesized Ruddlesden–Popper. The asterisks denote NdGaO₃ substrate peaks. **b** Resistivity measurements of the Nd₄Ni₃O₁₀ and Nd₄Ni₃O₈ films in (**a**). Data reproduced from ref. [14].

cyan and yellow, respectively. We observe an identical (but slightly rotated) structure on one side of the canted lattice near the top left of the image. Between the two, however, atomic distances in a subset of the positively canted region are better described by a near-90° rotation of the 3 × 3 rectangle. Even where the planar Ruddlesden–Popper structure is preserved, the in-plane spacings are shorter than the out-of-plane spacings, suggesting the internal oxygen lattice in this region differs from elsewhere in the film. Extracting the precise atomic structure of these defects is challenging given the higher degree of disorder in the surrounding lattice and the small total volume they comprise. While the observation of such competing reorientation likely does not strongly impact the macroscopic properties of the films studied here, it may inspire future efforts to decouple the various atomic sublattices in these or other layered materials.

These results suggest that to mitigate strain-induced extended defects upon reduction, the compressive strain in the reduced state should be minimized. NdGaO₃ is an appealing option as the compressive strain of the reduced compound is limited to 1.5%, compared to 3.2% on LaAlO₃ (Fig. 1). In Fig. 8, we present the reduction of the same Nd₄Ni₃O₁₀/NdGaO₃ film discussed in Fig. 3. Upon reduction, the

superlattice peaks in Fig. 8a shift toward positions consistent with the square-planar structure. Unlike the films on LaAlO₃ which partially relax upon reduction (Supplementary Fig. S12), we observe that Nd₄Ni₃O₈/NdGaO₃ is epitaxially strained to the substrate (Supplementary Fig. S22). As shown in Fig. 8b, the as-synthesized Nd₄Ni₃O₁₀ film exhibits a metal-to-metal transition at ~150 K, consistent with the bulk compound[80,86]. Grown instead on LaAlO₃, Nd₄Ni₃O₁₀ possesses a metal-to-insulator transition (Fig. 6e and refs. [46,47]). The reduced Nd₄Ni₃O₈/NdGaO₃ is metallic with a resistive upturn, similar to the infinite-layer nickelates[1,35] and Pr₄Ni₃O₈[12,82]. More details on the reduction of Nd₄Ni₃O₁₀/NdGaO₃ can be found in Supplementary Note 5. Thus, metallic and epitaxially strained layered square-planar nickelates can be stabilized on NdGaO₃.

Cross-sectional STEM images of the reduced Nd₄Ni₃O₈/NdGaO₃ film in Fig. 8 are presented in Fig. 9. The large-scale MAADF-STEM image in Fig. 9a shows Nd₄Ni₃O₈ ordering of fair crystalline quality, disordered regions with extended defects, as well as reduction-induced disorder at the surface. We observe the same effects in the HAADF-STEM images in Fig. 9b–d, which show regions of varying representative levels of crystalline order. All three regions (and

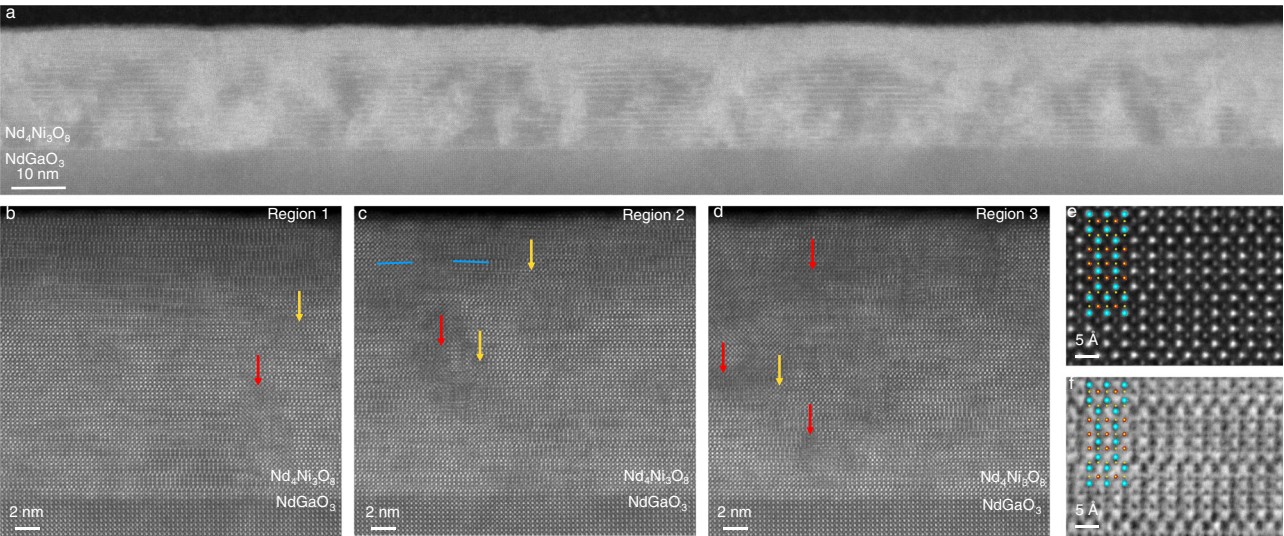

**Fig. 9 | Structural characterization of reduced the $Nd_4Ni_3O_8$/NdGaO$_3$ (110) films shown in Fig. 8. a** Large field-of-view MAADF-STEM image. **b–d** HAADF-STEM images of two separate regions with varying local densities of vertical rock salt faults. Blue lines highlight local *c*-axis canting, while red arrows mark disordered regions, and yellow arrows mark vertical rock salt faults without major amorphization. **e** HAADF and (**f**) ABF-STEM images of the same region. We provide the full field-of-view images in Supplementary Fig. S30.

especially Region 1 in Fig. 9b) show signs of reduced crystallinity near the film surface which are likely reduction-induced and not a result of STEM sample preparation (e.g. ion beam damage). In infinite-layer thin films, stabilizing capping layers of $SrTiO_3$ have proven an effective way to reduce similar surface degradation[37]. Near the top left of Region 2 in Fig. 9c, minor *c*-axis canting similar to that discussed in Fig. 7 can also be observed. All three regions show areas with significant disorder which may have nucleated near vertical rock salt faults in the as-synthesized film (see Figs. 3, 5b). Regions without amorphization but exhibiting vertical rock salt faults are highlighted with yellow arrows. In a relatively clean region of the film, HAADF- and ABF-STEM images in Fig. 9e, f show the expected $n = 3$ ordering and square-planar structure. Notably, the overall crystallinity of the $Nd_4Ni_3O_8$/NdGaO$_3$ film in Fig. 9 is higher than the crystallinity of the superconducting quintuple-layer nickelate[14], likely due to the challenges in stabilizing the higher order $n = 5$ phase. Both reduced $n = 3$ and $n = 5$ compounds on NdGaO$_3$, however, exhibit similar reduction-induced disorder, likely nucleating at vertical rock salt regions formed during the synthesis of the parent compound.

## Cation stoichiometry

Optimized $Nd_4Ni_3O_8$ ($n = 3$) and $Nd_6Ni_5O_{12}$ ($n = 5$) films on NdGaO$_3$ are metallic and superconducting, respectively[14]. However, the metallicity of these films is highly sensitive to as-synthesized structural differences and reduction conditions (Supplementary Fig. S21 and Supplementary Fig. S2 in ref. [14]). In the infinite-layer nickelates, cation stoichiometry strongly influences metallicity and superconductivity[7,37]. We thus investigate the role of cation stoichiometry in the metallicity of $Nd_4Ni_3O_8$ on NdGaO$_3$. In Fig. 10a we present XRD scans of five $Nd_4Ni_3O_{10}$ films with systematically varying neodymium content. The structural quality of the $Nd_4Ni_3O_{10}$ films deteriorates as the neodymium content is varied from the optimal value, evident by the broadness and reduced intensity of the 008 peak. Electrical transport measurements in Fig. 10b demonstrate that the resistivity of the $Nd_4Ni_3O_{10}$ films is relatively insensitive to cation stoichiometry, except for the 6% neodymium-rich film which exhibits higher resistivity than the rest of the films in the series. The decrease in resistive upturn temperature with increased off-composition is consistent with the decrease in metal–insulator transition temperature with neodymium-richness in NdNiO$_3$[61]. We provide more details

regarding the MBE synthesis and properties of these films in Supplementary Note 8.

Next we reduce the films shown in Fig. 10a and present the XRD scans in Fig. 10c. The three films within 3% of optimal stoichiometry exhibit all $Nd_4Ni_3O_8$ 00$l$ peaks, while the two films that deviate from optimal stoichiometry by 6% exhibit substantially diminished XRD peak intensity. These differences in structural quality are corroborated by the resistivity measurements in Fig. 10d. The films that deviate from optimal stoichiometry by 6% are insulating, while the films closer to optimal stoichiometry exhibit metallicity (more reduction trials are provided in Supplementary Figs. S16–S20). These results suggest that $Nd_4Ni_3O_8$ films can be metallic only if the cation stoichiometry lies within ~ 3% of the optimal value. It was similarly demonstrated that $Nd_{1-x}Sr_xNiO_2$ films are insulating if the cation stoichiometry is off by 10%[7]. However, we also observe metallicity in $Nd_4Ni_3O_8$ films that are structurally inferior to those shown in Fig. 10 (Supplementary Fig. S21c). The metallicity of reduced $Nd_4Ni_3O_8$ films is thus intricately dependent on a variety of structural factors including cation stoichiometry, oxygen content, and extended defect density.

## Discussion

In Fig. 6, we explore the role of rock salt layers in topotactic reductions. While the reduction of $Nd_4Ni_3O_{10}$ yields *c*-axis oriented $Nd_4Ni_3O_8$ with local *c*-axis canting, the reduction of NdNiO$_3$ primarily stabilizes *a*-axis oriented NdNiO$_2$. The primary distinction between NdNiO$_3$ and $Nd_4Ni_3O_{10}$ is the presence of rock salt layers in the Ruddlesden–Popper, which transform into fluorite-like NdO$_2$ layers through reduction (Fig. 1). Our results suggest that rock salt and fluorite-like spacer layers suppress *a*-axis reorientation and promote the deintercalation of apical oxygens along planes parallel to the spacer layers, as discussed in Supplementary Fig. S29. These spacer layers additionally facilitate the stabilization of the square-planar phase under much higher compressive strain than currently possible in infinite-layer systems. This capability is particularly appealing as the additional compressive strain has thus far enhanced $T_c$ in infinite-layer nickelates[10,11], although it is unclear to what extent this enhancement can be attributed to improved crystallinity. Additionally, oxygen diffusion is known to be anisotropic in Ruddlesden–Popper compounds, in contrast to perovskites[87,88]; the influence of rock salt layers on reduction kinetics, however, remains to be investigated.

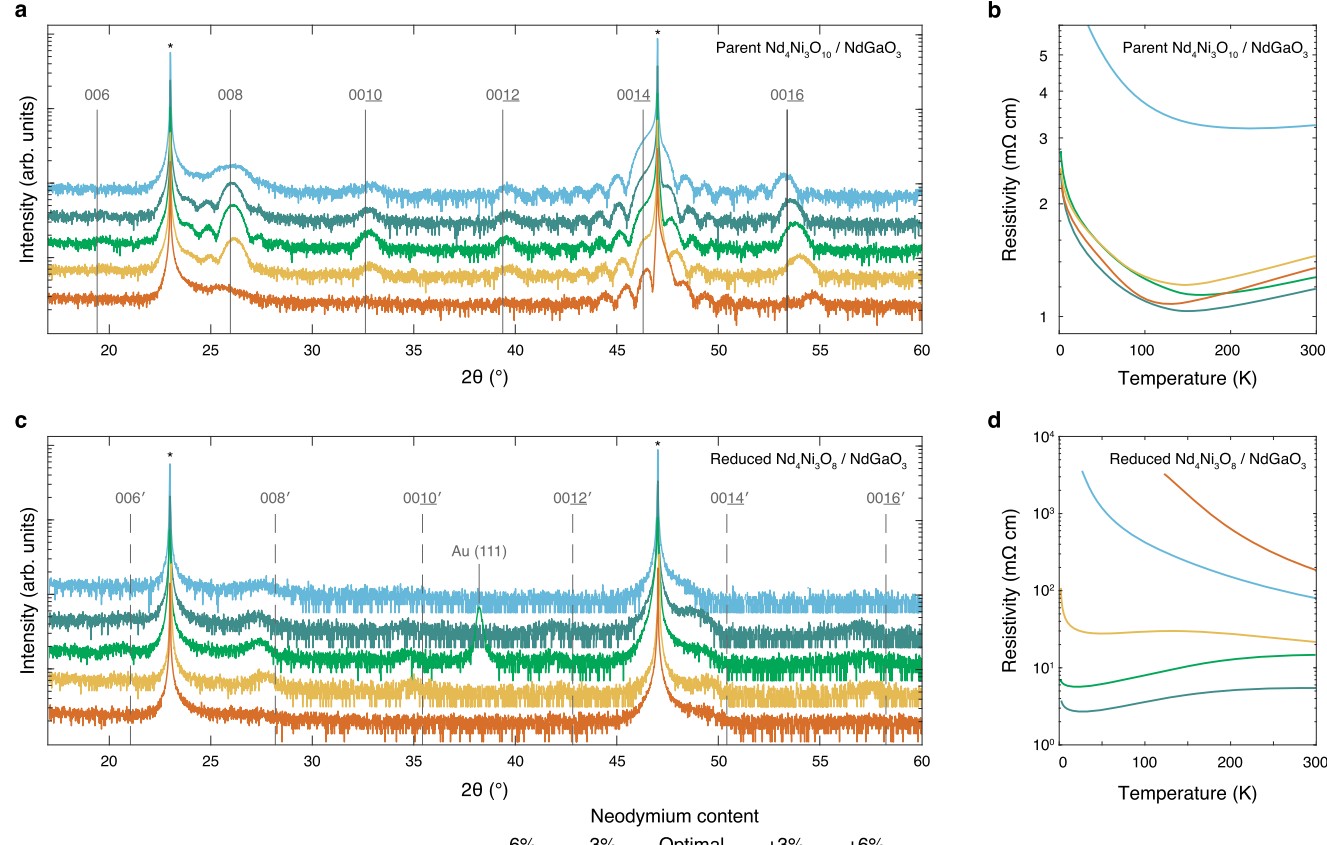

**Fig. 10 | Structural and electrical transport characterization of $Nd_4Ni_3O_{10}$ and $Nd_4Ni_3O_8$ films with varying neodymium content. a** XRD and (**b**) resistivity measurements of the parent $Nd_4Ni_3O_{10}$ films on $NdGaO_3$. **c** XRD and (**d**) resistivity measurements of the reduced $Nd_4Ni_3O_8$ films in (**a**). All films are ~11.0 nm and reduced for 3 hours at 290 °C. The vertical solid and dashed lines denote $00l$ peak positions of bulk $Nd_4Ni_3O_{10}$[95] and $Nd_4Ni_3O_8$[20], respectively. The primed indices distinguish the reduced layered square-planar phase from the as-synthesized Ruddlesden–Popper[20]. The asterisks denote $NdGaO_3$ substrate peaks. Additional reduction trials are provided in Supplementary Figs. S16–S20.

An additional issue raised by our work is the nature of the insulating and metallic states in reduced layered nickelates on $LaAlO_3$ and $NdGaO_3$, respectively. We speculate that in $Nd_4Ni_3O_8/LaAlO_3$, the compressive strain-induced extended defects shown in Fig. 7e preclude metallic transport through the film, despite the prevalence of high-crystallinity regions shown in Fig. 7b. Nevertheless, the potential role of the twinned structural domains in $LaAlO_3$ should be considered as well. The typical size of a twin domain in $LaAlO_3$ is ~1–100 μm[89]. The distance between $c$-axis canting defect regions in $Nd_4Ni_3O_8/LaAlO_3$, on the other hand, is ~100 nm, as shown in Fig. 7a. The length scale associated with $c$-axis canting defects is thus at least an order of magnitude smaller than the size of $LaAlO_3$ twin domains. This difference in length scale suggests the $c$-axis canting defects may form independently of the twin domains.

In summary, we have synthesized Ruddlesden–Popper $Nd_{n+1}Ni_nO_{3n+1}$ and layered square-planar $Nd_{n+1}Ni_nO_{2n+2}$ films under multiple strain states. We show that attempts to synthesize $Nd_4Ni_3O_{10}$ under 2.1% tensile strain on $SrTiO_3$ result in an extremely disordered structure with a high density of vertical rock salt faults, disqualifying these films from consideration for reduction. Synthesized under 0.8% tensile strain on $NdGaO_3$, $Nd_4Ni_3O_{10}$ exhibits horizontal rock salt ordering with a small density of vertical rock salt faults. We synthesize the highest quality $Nd_4Ni_3O_{10}$ films under 0.9% compressive strain on $LaAlO_3$, with few extended defects observed. Thus compared to perovskite nickelates, Ruddlesden–Popper films are more prone to the formation of tensile strain-induced extended defects because the horizontal rock salt layers can instead orient vertically to form strain-relieving rock salt faults. Therefore, minimizing the as-grown tensile

strain is crucial to decreasing the extended defect density in the reduced compounds, as was demonstrated in the infinite-layer nickelates[11]. Minimizing tensile strain is particularly crucial in the layered nickelates due to the increased propensity for the formation of strain-relieving extended defects compared to the infinite-layer nickelates.

We reduced the Ruddlesden–Popper nickelate films on $LaAlO_3$ and $NdGaO_3$ to the layered square-planar phase. In $Nd_4Ni_3O_8/LaAlO_3$, we observe $c$-axis canting diagonal defects interspersed between regions of high crystalline quality. All films reduced on $LaAlO_3$ are insulating, likely due to reduction-induced structural disorder. Reduced $NdNiO_2/LaAlO_3$ is also insulating but, in contrast to the layered nickelates, is primarily $a$-axis oriented. Horizontal rock salt layers in the Ruddlesden–Popper structure thus suppress $c$-axis reorientation during reduction under high compressive strain. Reduced $Nd_4Ni_3O_8$ films on $NdGaO_3$, on the other hand, demonstrate high-quality layered square-planar ordering as well as regions with extended defects such as vertical rock salt faults. Despite the presence of these extended defects, $Nd_4Ni_3O_8$ ($n = 3$) and $Nd_6Ni_5O_{12}$ ($n = 5$) films on $NdGaO_3$ are metallic and superconducting, respectively[14]. The metallic state in $Nd_4Ni_3O_8/NdGaO_3$, however, can only be stabilized if the neodymium content lies within ~3% of the optimal quantity. The competing requirements for the MBE synthesis of the parent compound and reduction to the square-planar phase are thus met by synthesizing the parent compound under tensile strain on $NdGaO_3$ to accommodate the large increase in the in-plane lattice parameter upon reduction, at the cost of forming strain-induced vertical rock salt faults in the as-synthesized film.

Through our systematic study across multiple substrates, we establish a method to synthesize layered square-planar nickelate thin films and set limits on the ability to strain-engineer these compounds. Furthermore, our DFT calculations demonstrate that $Nd_4Ni_3O_8$, if optimized to a metallic state on $LaAlO_3$ and $SrTiO_3$, could be a platform to study the role of the charge-transfer energy and rare-earth $5d$ states in nickelate superconductivity. This work provides a comprehensive starting point from which to launch future investigations into the role of epitaxial strain, dimensionality, and chemical doping in nickelate superconductivity.

## Methods

### Molecular beam epitaxy synthesis and $CaH_2$ reduction

We employ ozone-assisted MBE to synthesize the Ruddlesden–Popper nickelates on $LaAlO_3$ (001), $NdGaO_3$ (110), and $SrTiO_3$ (001). To calibrate the nickel and neodymium elemental fluxes, we synthesize NiO on MgO (001) and $Nd_2O_3$ on yttria-stabilized zirconia (YSZ (111)), then measure the film thickness via x-ray reflectivity[90]. Next, we synthesize $NdNiO_3/LaAlO_3$ (001) and use the $c$-axis lattice constant and film thickness to refine the Nd/Ni ratio and monolayer dose, respectively[7]. Using the optimized neodymium and nickel shutter times from the synthesis of $NdNiO_3/LaAlO_3$, we synthesize the Ruddlesden–Popper nickelates via monolayer shuttering. Both $NdNiO_3$ and Ruddlesden–Popper nickelates are synthesized with ~1.0e−6 Torr distilled ozone (Heeg Vacuum Engineering) and 500–600 °C manipulator temperature. The MBE synthesis conditions and calibration scheme are described in refs. [14,47]; similar techniques were also used in refs. [45,46].

The Ruddlesden–Popper films were reduced to the layered square-planar phase using $CaH_2$ topotactic reduction. The as-synthesized films are cut into ~2.5 × 5-mm$^2$ pieces, wrapped in aluminum foil (All-Foils), then inserted into borosilicate tubes (Chemglass Life Sciences) with ~0.1 g of $CaH_2$ pellets (>92%, Alfa Aesar). The borosilicate tube is sealed at <0.5 mTorr using a small turbomolecular. The sealed glass ampoule is heated in a convection oven (Heratherm, Thermo Fisher Scientific) for several hours at ~290 °C, with a 10 °C min$^{-1}$ heating rate. After reduction, the film is rinsed in 2-butanone and isopropanol to remove $CaH_2$ residue. Similar methods are commonly used elsewhere[7,14,37].

### Structural characterization

X-ray diffraction (XRD) measurements were performed using a Malvern Panalytical Empyrean diffractometer with Cu K$\alpha_1$ ($\lambda = 1.5406$ Å) radiation. Reciprocal space maps (RSMs) were taken with a PIXcel3D area detector. Lattice constants were calculated using Nelson-Riley fits of the superlattice peak positions[91].

Cross-sectional STEM specimens were prepared using the standard focused ion beam (FIB) lift-out process on a Thermo Scientific Helios G4 UX FIB or an FEI Helios 660. High-angle annular dark-field (HAADF-) and medium-angle annular dark field (MAADF-) STEM images were acquired on an aberration-corrected Thermo Fisher Scientific Spectra 300 X-CFEG operated at 300 kV with a probe convergence semi-angle of 30 mrad and inner collection angles of 66 mrad (HAADF) or 17 mrad (MAADF). Annular bright field (ABF)-STEM images were acquired on an aberration-corrected 300 kV FEI Titan Themis with a probe convergence semi-angle of 21.4 mrad and 12 mrad inner collection angle. Additional HAADF-STEM measurements were performed on a Thermo Fisher Titan Themis Z G3 operated at 200 kV with probe convergence semi-angle of 19 mrad and inner collection angle of 71 mrad.

Lattice-scale disorder in the as-synthesized films is visualized by extracting modulations in the (101) and ($\bar{1}$01) pseudocubic lattice fringes using the phase lock-in method described in refs. [92,93] and implemented by the Python analysis package publicly available at https://doi.org/10.34863/amcp-4s12. In particular, vertical and horizontal rock salt planes or rock salt faults appear as an apparent strong

local compressive strain in the pseudocubic lattice fringes. The original HAADF-STEM images used for analysis and the raw output strain maps are provided in Supplementary Fig. S26. Furthermore, regions within each Ruddlesden–Popper layer show small negative strain values: this is due to the choice of the reference lattice spacing, which is based on the average image in this Fourier-based technique and is not reflective of real elastic strain in the atomic lattice. To highlight the Ruddlesden–Popper faults, we therefore include only positive strain values in the overlays shown in Fig. 5 with the full maps shown in Supplementary Fig. S27. The transparency of the strain map overlays in Fig. 5 follow the local magnitude of the strain.

The map of local $c$-axis orientation in Fig. 7 is generated with the local wave fitting analysis described in ref. [85] using the 002 pseudocubic lattice fringes, cropped windows of 24 × 24 pixels, and a window step size of 12 pixels equivalent to ~ 0.16 nm. The map displayed in Fig. 7 is additionally smoothed by a Gaussian kernel with $\sigma = 5$ corresponding to a distance of about 5 pseudocubic unit cells; the original STEM image, raw wave fitting output, and smoothed result are provided in Supplementary Fig. S27.

Electron energy loss spectroscopy (EELS) measurements were carried out on the Thermo Fisher Scientific Spectra 300 X-FEG equipped with a Gatan Continuum spectrometer and camera. Spectrum images of the films on $LaAlO_3$ and $NdGaO_3$ were acquired with a spectrometer dispersion of 0.3 eV per channel. Spectrum images of the film on $SrTiO_3$ were acquired operating in DualEELS mode with a spectrometer dispersion of 0.15 eV per channel. The accelerating voltage was 300 (120) kV for measurements of the films on $SrTiO_3$ and $NdGaO_3$ ($LaAlO_3$). Due to their overlapping EELS edges, two-dimensional concentration maps of the La-$M_{4,5}$ and Ni-$L_{2,3}$ edges are determined by non-negative least squares (NNLS) fit to the weighted sum of reference components for each edge taken from the substrate (La-$M_{4,5}$) and film (Ni-$L_{2,3}$) regions (Supplementary Fig. S25).

### Electrical transport

Electrical transport data were primarily taken using a Quantum Design Physical Property Measurements System equipped with a 9 T magnet. Hall bars were patterned with Cr (5 nm)/Au (100 nm) contacts using shadow masks which were then defined using a diamond scribe. Resistivity data were taken down to 1.8K at ~3 °C/min using an AC lock-in amplifier. Hall coefficients were determined from linear fits of antisymmetrized field sweeps up to 9T. All field sweeps were taken upon warming.

Several electrical transport measurements in the Supplementary Materials were taken using a home-built electrical 'dipstick probe' compatible with a helium dewar. Indium contacts were soldered on the four corners of each film in a Van der Pauw configuration. AC transport measurements were taken at 17.777 Hz using SR830 lock-in amplifiers. The voltage and current were measured simultaneously to determine the resistance.

### X-ray absorption spectroscopy

X-ray absorption spectroscopy (XAS) was performed at the Advanced Light Source, Lawrence Berkeley National Lab, at Beamline 6.3.1. The spectra were taken in the total electron yield mode at 300K with linear horizontally polarized light. The film was oriented either normal ($I_x$) or at 30° grazing incidence ($I_z$) to the beam. For the signal acquired at grazing incidence, a geometric correction factor was applied[94]. The spectra are normalized to the incident x-ray flux and scaled to the same intensity at energies just below the absorption edge. Every spectrum we present is an average over eight pairs of spectra measured in normal ($I_x$) and grazing ($I_z$) x-ray incidence angle.

## Data availability

Source data for x-ray diffraction and electrical transport in Figs. 2–10 and high-resolution STEM images contained within the main text are

provided in the Source Data file. Additional data which support the findings of this study are available from the corresponding authors upon reasonable request. Source data are provided with this paper.

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

## Acknowledgements

D.F.S. and B.H.G. contributed equally to this work. We thank Jennifer E. Hoffman and Michael Hayward for fruitful discussions. We thank H. Hijazi at the Rutgers University Laboratory of Surface Modification for assistance in Rutherford backscattering spectrometry. This project was primarily supported by the US Department of Energy, Office of Basic Energy Sciences, Division of Materials Sciences and Engineering, under Award No. DE-SC0021925. Materials growth and electron microscopy were supported in part by the Platform for the Accelerated Realization, Analysis, and Discovery of Interface Materials (PARADIM) under NSF Cooperative Agreement No. DMR-2039380. Electron microscopy was primarily performed at the Cornell Center for Materials Research (CCMR) Shared Facilities, which are supported by the NSF MRSEC Program (No. DMR-1719875). Additional electron microscopy was performed using the MIT.nano facilities at the Massachusetts Institute of Technology and at Harvard University's Center for Nanoscale Systems (CNS), a member of the National Nanotechnology Coordinated Infrastructure Network (NNCI), supported by the NSF division of Electrical, Communications and Cyber Systems (ECCS) under NSF Grant No. 2025158. This work made use of the Advanced Light Source, a U.S. DOE Office of Science User Facility under contract No. DE-AC02-05CH11231. D.F.S. acknowledges support from the US Department of Energy, Office of Basic Energy Sciences, Division of Materials Sciences and Engineering, under award no. DE-SC0021925 and from the NSF Graduate Research Fellowship no. DGE-1745303. B.H.G. and L.F.K acknowledge support from PARADIM and the Packard Foundation. G.A.P. acknowledges support from the Paul & Daisy Soros Fellowship for New Americans and from the NSF Graduate Research Fellowship Grant No. DGE-1745303. Q.S. was supported by the STC Center for Integrated Quantum Materials, NSF Grant No. DMR-1231319. H.E.S. and I.E. were supported by the Rowland Institute at Harvard. J. A. Mason acknowledges support from the Arnold and Mabel Beckman Foundation through a Beckman Young Investigator grant. H.L. and A.S.B. acknowledge support from NSF Grant No. DMR-2045826. J.A. Mundy acknowledges support from the Packard Foundation and the Gordon and Betty Moore Foundation's EPiQS Initiative, grant GBMF6760.

## Author contributions

D.F.S., G.A.P., Q.S., C.M.B., and J.A.Mu. synthesized the thin films with assistance from H.P. D.F.S., G.A.P., A.T., N.K.T., and S.D. conducted the reductions with guidance from J.A.Ma. D.F.S and G.A.P. performed the transport measurements. B.H.G., H.E.S., I.E.B., and L.F.K characterized the samples with scanning transmission electron microscopy. B.H.G. and L.F.K. carried out the STEM image analysis. G.A.P., N.K.T., D.F.S., and A.T.N. performed x-ray absorption spectroscopy. H.L., M.J., and A.S.B. performed DFT-based electronic structure calculations. J.A.Mu. conceived and guided the study. D.F.S., B.H.G., and J.A.Mu. wrote the manuscript with discussion and contributions from all authors.

## Competing interests

The authors declare no competing interests.
