## [Peer Review File · Nature Communications]

Limits to the strain-engineering of layered square-planar nickelate thin filmsREVIEWER COMMENTS

Reviewer #1 (Remarks to the Author):

In this work, Segedin et al tries to claim the effect of strain on the synthesis of square-planar nickelate thin films, which are associated with the emergent Ni-based superconducting oxides. Unfortunately, the main claims are rather scattered and none of the claims is supported by systematic experiments. Lots of important factors and experimental tuning parameters during the chemical reduction were not considered at all.

For example: the authors found that for parent RP phase compounds, the thin films have the highest quality if growing on LAO substrates. However, after reduction, the films become insulating. In contrast, only parent films with moderate defects grown on NdGaO₃ substrates are good for achieving superconductivity upon chemical reduction. This is the only positive result but has been published in the authors' previous work [Ref. 14].

Indeed, the above two facts are very confusing. The authors seem to emphasize that defects are important for chemical reduction in addition to different strain conditions. But it is definitely inconsistent with common sense. The reason for this inconsistent situation is that the authors failed to consider the key factors in chemical reduction, thin film thickness that is related to the diffusion length of O under reducing conditions, temperature for reduction, the time of chemical reduction, and the volume of CaH₂ powder and other chemical parameters. For instance, the recent experimental study on chemical reduction [Puphal et al., Sci. Adv. 7, eabl8091 (2021)] shows as long as one takes sufficiently long time, metallic transport with the 112 phase could be achieved for bulk materials rather than thin films.

Overall, the chemical reduction process might be also related to the strain situation, defects situation of the parent thin films, but without considering the most important experimental parameters for this chemical reaction process. talking about strain and defects in parent films is rather trivial, invalid and misleading.

More importantly, the manuscript did not contain any striking point suitable for nature communications. I suggest that the authors shall focus on more important basic experimental tuning parameters first before directly jumping to strain and physics and then organize it as a systematic work, which may then be suitable for a specialized journal such as Physical Review Materials for Journal of Applied Physics.

Reviewer #2 (Remarks to the Author):

In the manuscript by Segedin et al., the authors reported the strain engineering of layered square-planar nickelate thin films. By synthesizing nickelate films on three different substrates, the authors explored the impact of strain on the crystalline quality of both the precursor Ruddlesden-Popper (RP) and reduced square-planar phases by XRD and STEM. Their results suggest that the crystalline quality of the RP films decreases as the tensile strain increases, resulting in best quality films on LaAlO₃ and worst quality films with disordered structure on SrTiO₃. As the reduction of the nickelate films experience a large expansion of the in-plane lattice constants, the films grown on NaGaO₃ has better lattice mismatch to accommodate such expansion and exhibit fair crystalline quality both before

and after the reduction process. The authors also found that the rock-salt structures in RP films help stabilize the c-axis orientation of the reduced phase in the reduction process. The layered $\text{Nd}_{n+1}\text{Ni}_n\text{O}_{2n+2}$ series are of great interests since they are predicted to exhibit more cuprate-like electronic structure and enhance T_c . However, the synthesis of such superconducting nickelate films has been an extremely challenging task. This work provides valuable information about the synthesis of high-quality films of this series. In the following, I have a few questions/concerns that I would like the authors to address.

1) It is clear that these three types of substrates used in this work can provide different strain states for the nickelate films, which is the reason that the authors attribute the observed phenomena to the strain effects. However, there are also some other distinct differences between these substrates. Especially, the SrTiO_3 has a non-polar structure, in contrast to the polar structure of other substrates. As the rock-salt layer has the tendency to increase the polar discontinuity, it could be harder to form compared to the perovskite NdNiO_3 . The rock-salt structure in RP nickelate films may be able to be stabilized on SrTiO_3 by properly engineering the interface structure, i.e. with a polar buffer layer. Nonetheless, it is important to rule out the polar discontinuity effect in order to make solid discussions on the strain effects.

2) In the manuscript, the author mainly discussed the strain effects based on the experimental observations. It would be helpful if the authors can provide theoretical simulations to support their arguments.

3) Is there any reason why $\text{Nd}_4\text{Ni}_3\text{O}_{10}/\text{LAO}$ show substantially better crystallinity than $\text{Nd}_4\text{Ni}_3\text{O}_{10}/\text{NGO}$ as the absolute values of the lattice mismatch are nearly the same ($\sim 1\%$) for $\text{Nd}_4\text{Ni}_3\text{O}_{10}$ grown on these two substrates. Could the authors comment on this point?

4) The XRD patterns indicate that the stoichiometry in present RP films grown on different substrates deviates from the ideal values (the Δc is not identical for two adjacent 00l diffraction peaks). Therefore, I am curious about what role does the stoichiometry play in terms of structural and transport properties, because it seems the metallic behavior is more robust for films grown on NGO than LAO, although the disorder level is very similar for these two cases.

Reviewer #3 (Remarks to the Author):

This is a beautiful study of the structural and transport properties of nickelate RP films and their reduced variants. Currently, this group is the only one in the world who has been able to make and study such reduced RP films. As such, I strongly support the publication of this paper in Nature Communications.

What the paper lacks, though, is a comparison to previous studies of reduced bulk materials. The first paper I would like to point out is Retoux et al (J. Solid State Chem. 140, 307 (1998)) who studied defects in bulk $\text{Nd}_4\text{Ni}_3\text{O}_{10}$. They primarily found stacking faults with reduced n ($n=1$ and 2), but occasionally found ones of higher n (like $n=6$). A brief comparison to this paper would be in order.

Second, I looked through the literature and found all the papers I could where transport was measured on bulk $\text{Nd}_4\text{Ni}_3\text{O}_{10}$. This involved four published papers along with a slide I had from a

conference proceedings. Most of these found insulating behavior like the authors find for Nd₄₃₈ on LAO. But one of these papers (Miyatake et al.) found results analogous to what the authors find for Nd₄₃₈ on NGO. These materials were processed with sulfur, which they claim acts as an oxygen getter. This implies that this difference might be due to stoichiometry and thus the Ni valence. It could well be that the density wave state in the 438 materials that is associated with insulating behavior only exists in a narrow doping range, similar to what is seen for some of the charge ordered states in layered manganites.

In that context, the authors early on in the paper discuss Nd enriched samples and associated insertion of extra NdO layers, but this argumentation did not seem to be extended to the reduced materials. Imagine that one inserts x layers per formula unit of NdO. Then the resulting nickel valence will be reduced by $x/3$. That is, one would effectively be electron doping relative to $1/3$. Perhaps such a reduced Ni valence relative to $1/3$ is associated with metallic behavior? This is worth some thought.

**Response to reviewers for “Limits to the strain engineering of
layered square-planar nickelate thin films” (NCOMMS-22-29067)**

(Dated: December 1, 2022)

REVIEWER #1 (REMARKS TO THE AUTHOR):

In this work, Segedin et al tries to claim the effect of strain on the synthesis of square-planar nickelate thin films, which are associated with the emergent Ni-based superconducting oxides. Unfortunately, the main claims are rather scattered and none of the claims is supported by systematic experiments. Lots of important factors and experimental tuning parameters during the chemical reduction were not considered at all.

Our response: We appreciate the reviewer’s consideration of our work. We agree with the reviewer that experimental tuning parameters during reduction are critical to the process. Indeed, Supplemental Figures S1-4 in our originally submitted manuscript documented the impact of reduction temperature and time on the resulting structure and electrical properties of our $\text{Nd}_4\text{Ni}_3\text{O}_8$ films. We performed extensive testing of the reduction conditions with the synthesis of 30+ $\text{Nd}_4\text{Ni}_3\text{O}_{10}$ thin films and 50+ separate reduction experiments to the $\text{Nd}_4\text{Ni}_3\text{O}_8$ phase, as summarized in the original Supplemental Figures S1-4. We also synthesized $R\text{NiO}_3$ thin films from which we performed 50+ reductions to construct the infinite-layer $R\text{NiO}_2$ phase to benchmark our studies.

As documented below, our main conclusions from those studies are that a) the optimal reduction conditions are the same for films on LaAlO_3 and NdGaO_3 , and b) that the optimal reduction conditions (~ 3 hours at 290°C) for our $\text{Nd}_4\text{Ni}_3\text{O}_8$ are the same as those reported for $R\text{NiO}_2$ films, which are well documented by the infinite-layer nickelate community^{1,2}. Throughout this study, we used these optimal reduction conditions, controlling for the reduction temperature and time, as well as the film thickness and CaH_2 powder volume. Here, we found a striking difference in $\text{Nd}_4\text{Ni}_3\text{O}_8$ electrical properties and structure as a function of substrate strain which could not be explained by variations in the reduction process. Below we document our response to each of the variables listed by the reviewer:

– Reduction temperature/time: The reviewer is correct that the temperature at which the CaH_2 reduction is performed and the time of the reaction are critical to establishing the square-planar phase. If the reaction is run at insufficient temperature or for insufficient time, there will be an incomplete reduction to the targeted square-planar phase, as demonstrated in Fig. S4 of our original manuscript. We have performed additional reductions and transport measurements to show that the precise reduction conditions have a minor effect on the

transport properties. In Fig. R1 we demonstrate that incremental reductions of a $\text{Nd}_4\text{Ni}_3\text{O}_{10}$ / NdGaO_3 film have a minimal effect on both the resistivity and hall coefficient of the sample (figure added to supplement):

FIG. R1: X-ray and transport characterization of an incrementally reduced $\text{Nd}_4\text{Ni}_3\text{O}_{10}$ / NdGaO_3 film. (a) XRD scans of an as-grown film (red) then consecutively reduced for 3 hours at 290°C (yellow), 2 hours at 290°C (blue), and 6 hours at 300°C (green). (b) Resistivity versus temperature and (c) hall measurements of the reduced samples in (a).

Furthermore, we emphasize in Figs. R3 (added to supplement), R4, R5 and R2 that we have systematically explored a wide variety of reduction conditions, but find that 3 hours at 290°C most reliably yields metallic transport. Indeed, our optimal reduction temperature and time (3 hours at 290°C) is consistent with the reduction conditions used by most other groups in the infinite-layer nickelate community¹⁻³. In Fig. R2 we share our reduction

optimizations (from Fall 2019) on perovskite PrNiO_3 / SrTiO_3 in an attempt to reproduce the Stanford group’s initial report of superconductivity. With systematic experiments like those shown in Fig. R2, we narrowed the optimal reduction conditions by tracking the structural evolution (as well as the transport properties) from the perovskite to the infinite-layer phase. Once we optimized the perovskite reductions, we confirmed that the optimal reduction conditions also match those for the layered nickelates. We consistently this optimal reduction condition (~ 3 hours at $290\text{-}300^\circ\text{C}$) throughout our study.

FIG. R2: Systematic reductions optimizations performed on a single 10 nm PrNiO_3 / SrTiO_3 film. The top row consists of 1 hour reductions at varying temperatures, while the bottom row consists of 3 hours reductions at varying temperatures. Phase coexistence evident by peak splitting is apparent for 1 and 3 hour reductions at 280°C . The 3 hour at 290°C reduction, however, exhibits single-phase PrNiO_2 .

In Fig. R3 (added to supplement), we show a series of reductions on the $\text{Nd}_4\text{Ni}_3\text{O}_{10}$ sample of optimal stoichiometry discussed in our new section on cation stoichiometry (see Fig. R17). We demonstrate that lower temperature (240°C) reductions can yield the reduced phase,

structurally. However, metallicity can only be achieved at higher temperatures (290°C, see Fig. R17 for transport on this sample). In fact, the top sample in the figure (reduced for 3 hours at 290°C, with 80kΩ surface resistance) is the only metallic sample out of the series. We furthermore emphasize that, as reported by other groups, chemical reductions are difficult to reproduce and can be stoichastic from reduction to reduction (see top three reductions at 290°C in Fig. R3). Nevertheless, the stochasticity in our reductions is low enough for us to reliably reproduce metallic Nd₄Ni₃O₈ samples. Figures R4 and R5 corroborate the above discussion for an additional two Nd₄Ni₃O₁₀ / NdGaO₃ samples.

FIG. R3: Reductions of the ‘optimal’ Nd₄Ni₃O₁₀ / NdGaO₃ sample in Fig. R17 and Supplementary Figures R19 and R18. Reduction conditions (hours and temperature in degrees celsius) and surface resistances (measured by multimeter) are shown in the legend. The peak at ~37° is the Au (111) peak. We note that metallic transport is only achieved by the top sample with an 80kΩ surface resistance after reduction for 3 hours at 290°C.

FIG. R4: Reductions of a $\text{Nd}_4\text{Ni}_3\text{O}_{10}$ / NdGaO_3 sample. Reduction conditions (hours and temperature in degrees celsius) and surface resistances (measured by multimeter) are shown in the legend. The peak at $\sim 37^\circ$ is the Au (111) peak. We note that although the lower temperature (less than 250°C) reductions yield XRD that appears fully reduced, the surface resistance is very high. This is likely indicative of slight oxygen deficiencies in the sample. Thus higher temperature is required to fully remove the oxygen.

FIG. R5: Reductions of a $\text{Nd}_4\text{Ni}_3\text{O}_{10}$ / NdGaO_3 sample. Reduction conditions (hours and temperature in degrees celsius) and surface resistances (measured by multimeter) are shown in the legend. Here we highlight the second XRD scan from the bottom (reduced for 6 hours at 220°C), which shows phase coexistence between the as-synthesized and reduced structures. The highest quality structure and lowest resistance here is achieved with the sample shown at the top, reduced for 2 hours at 280°C .

– Volume of the CaH₂ powder: Our studies have not found this to be a significant variable in the reduction process. In contrast to powder and bulk crystal studies in which the CaH₂ powder is added as a stoichiometric quantity, the CaH₂ powder has $\sim 10^7$ of the molar quantity of our thin films. Thus it is perhaps unsurprising that the volume of the CaH₂ added does not seem to have a major impact on the reaction process. Furthermore, we keep the mass of the CaH₂ powder constant at 0.1 g throughout all of our reductions.

– Film thickness: While variations in thickness cannot explain the results in our original manuscript, we nevertheless appreciate the reviewer’s comment that this could be an additional interesting point to consider. In our originally submitted manuscript, all films shown were ~ 10 -25 nm. Furthermore, as we are interested in the strain-dependent properties of our thin films, we were restricted to films which were below the critical thickness for strain-relaxation. In response to the reviewer’s comment, we have performed a follow-up study in which we synthesized two films with different thicknesses on NdGaO₃. We demonstrate in Fig. R6 (added to supplement) that two samples with dramatically different thicknesses (11.0 nm and 25.5 nm) can both be reduced in 3 hours to a metallic state with very similar resistivity. Furthermore, we emphasize that we have **not once** achieved a metallic reduced nickelate on LaAlO₃ (NdNiO₂, Nd₄Ni₈, and Nd₆Ni₅O₁₂ were all attempted). Across the dozens of samples attempted (some of which are surveyed in our original manuscript in Figs. S1-S6), the thickness has varied from 10 nm to 30 nm.

FIG. R6: Thickness-dependent reductions of Nd₄Ni₃O₁₀ / NdGaO₃. (a) XRD scans of as-grown and reduced films of 25.5 nm and 11.0 nm thicknesses. The 25.5 nm and 11.0 nm films were reduced for 3 hours at 300°C and 3 hours at 290°C, respectively. (b) Resistivity versus temperature measurements of the reduced films in (a).

We absolutely agree with the Reviewer that thickness is an important quantity to keep track of and are aware that the infinite-layer community has learned that maximal sample uniformity is achieved with thinner samples (~ 10 nm)⁴. In fact, we have recently started to grow mostly ~ 10 nm films for reductions. However, small variations in thickness cannot explain the **strain-dependent** phenomena we present in our manuscript.

– Stoichiometry of the as-grown film: We appreciate both reviewer 2 and reviewer 3’s inquiries about the cation stoichiometry. Indeed, this is a critical variable and impacts the electrical properties of the as-grown film and has a dramatic impact on the electrical properties of the reduced films. In response to the reviewers’ comments, we synthesized a series of $\text{Nd}_4\text{Ni}_3\text{O}_{10}$ / NdGaO_3 films with systematically varying neodymium content. The results are presented in Figs. R17, R18, R19, and R20 at the end of this document. In summary: we have shown that the metallicity of the reduced $\text{Nd}_4\text{Ni}_3\text{O}_8$ is exquisitely sensitive to the cation stoichiometry. In fact, metallicity in $\text{Nd}_4\text{Ni}_3\text{O}_8$ can only be achieved if the as-grown $\text{Nd}_4\text{Ni}_3\text{O}_{10}$ compound is within 3% of optimal neodymium content. This is a crucial piece of new information for any group attempting to synthesize metallic, let alone, superconducting layered nickelates.

We have added expanded discussions on the effect of cation stoichiometry to the Main Text and as Supplemental Note 8, as included above / below (pages 36-42 of this response letter).

For example: the authors found that for parent RP phase compounds, the thin films have the highest quality if growing on LAO substrates. However, after reduction, the films become insulating. In contrast, only parent films with moderate defects grown on NdGaO_3 substrates are good for achieving superconductivity upon chemical reduction. This is the only positive result but has been published in the authors’ previous work [Ref. 14].

*Indeed, the above two facts are very confusing. The authors seem to emphasize that defects are important for chemical reduction in addition to different strain conditions. But it is definitely inconsistent with common sense. The reason for this inconsistent situation is that the authors failed to consider the key factors in chemical reduction, thin film thickness that is related to the diffusion length of O under reducing conditions, temperature for reduction, the time of chemical reduction, and the volume of CaH_2 powder and other chemical parameters. For instance, the recent experimental study on chemical reduction [Puphal et al., *Sci. Adv.* 7, eabl8091 (2021)] shows as long as one takes sufficiently long time, metallic transport with the 112 phase could be achieved for bulk materials rather than thin films. **Our***

response: As documented above we have indeed carefully optimized the key factors in the

reduction process (see Supplemental Materials of our original manuscript) and, more critically, kept these parameters of the reduction (film thickness, reduction temperature/time, CaH_2 volume) constant across all of our presented data.

We first discuss the paper the Reviewer cites [Puphal et al., *Sci. Adv.* 7, eabl8091 (2021)] point-by-point to show the paper is entirely consistent with our observations. Puphal et al. state: ‘We found that after 1 day of reduction at 300°C , the crystals transformed into an intermediate phase (likely the $\text{La}_{1-x}\text{Ca}_x\text{NiO}_{2.5}$ phase). The subsequent reduction to the infinite-layer phase can be accomplished with a substantially longer reduction time of approximately 2 weeks.’ The Reviewer is correct that bulk crystals of both the LaNiO_3 and the layered $R_4\text{Ni}_3\text{O}_{10}$ compounds have been reduced to the square planar form. The reaction times are, however, much longer for bulk crystals due to the larger volume, and thus much lower surface area to volume ratio than the film samples. Instead of one day at 300°C like Puphal et al., we see intermediate reduction after 6 hours at 220°C in Fig. R5, and 3 hours at 280°C in Fig. S4 of our originally submitted manuscript – full reductions are also shown for both of these cases after either longer reduction time or higher reduction temperature was used.

Puphal et al. also discuss stochasticity: ‘The optimal duration of the reduction is individual for each crystal and depends on details, such as crystal size, shape, and the Ca substitution level.’ Their reported stochasticity is consistent with the stochasticity evident in Figures R3, R4, R5, and Fig. S7 of our originally submitted manuscript. In fact, sample details like strain and cation stoichiometry are much more important for the properties of the reduced compound than the precise reduction conditions (as shown in Figs. R6 and R17).

Furthermore, Puphal et al. show that complete reduction can be achieved after sufficiently long reduction, but only minor changes (and sample degradation) occur for reduction after reaching the reduced phase: ‘Overall, we find that an extension of the 2-week time period by a few days did not induce obvious changes in crystals with 0.06×0.16 and sizes between 75 and $200 \mu\text{m}$ ’. The fact that only minor changes occur after reducing for long periods is illustrated in Fig. S7(a) of our originally submitted manuscript, where metallicity in $\text{Nd}_4\text{Ni}_3\text{O}_8 / \text{NdGaO}_3$ is evident after reduction for 3 hours at 280°C and even after 24 hours at 300°C . To demonstrate that our LaAlO_3 -grown samples are not insulating due to insufficient reduction, we have performed a long reduction on a already-reduced $\text{Nd}_4\text{Ni}_3\text{O}_8$

/ LaAlO_3 film (Fig. R7, added to supplement). Evidently, the structure deteriorates after an addition 12 hours of reduction at 290°C (Fig. R7(a)). However, the resistivity in Fig. R7(b) is unchanged after the additional reduction time. We perform the same experiment on the 3% neodymium rich/poor samples discussed in the context of cation stoichiometry in Fig. R17, and again see that insulating behavior persists after an additional 12 hours of reduction (Fig. R8, added to supplement). We also note that our electron microscopy, the x-ray diffraction, and XAS data (Fig. S13) confirm that we have indeed reached the square-planar form for samples on LaAlO_3 but the samples remain insulating.

FIG. R7: (a) XRD scans of (from bottom to top) a $\text{Nd}_4\text{Ni}_3\text{O}_{10}$ / LaAlO_3 film as-grown, reduced for 3 hours at 290°C , and subsequently reduced for an additional 12 hours at 290°C . (b) Resistivity versus temperature measurements of the reduced films in (a). The reduced film peak intensities after 12 hours of reduction in (a) suggest structural degradation. Interestingly, the resistivity versus temperature behavior is nearly identical for the two reduction conditions.

FIG. R8: (a) XRD and (b) transport measurements of the 3% neodymium-rich sample in main text figure R17. (c) XRD and (d) transport measurements of the 3% neodymium-poor sample in main text figure R17. The samples were reduced for (i) 3 hours at 290°C, and consecutively for (ii) 3 hours at 290°C, then (iii) 12 hours at 290°C. The trials here demonstrate the robust nature of the insulating state in these films.

Overall, the chemical reduction process might be also related to the strain situation, defects situation of the parent thin films, but without considering the most important experimental parameters for this chemical reaction process, talking about strain and defects in parent films is rather trivial, invalid and misleading.

More importantly, the manuscript did not contain any striking point suitable for nature communications. I suggest that the authors shall focus on more important basic experimental tuning parameters first before directly jumping to strain and physics and then organize it as a systematic work, which may then be suitable for a specialized journal such as Physical Review Materials for Journal of Applied Physics.

Our response:

We agree with the reviewer that this is fascinating – and perhaps seemingly counter-intuitive. In both the infinite-layer compounds and our superconducting $\text{Nd}_6\text{Ni}_5\text{O}_{12}$, there is a delicate balance between the competing requirements for the synthesis of both the oxidized compounds and the reduced phase. Indeed, recent work on $R\text{NiO}_2$ has demonstrated that the superconducting T_c can be raised from ~ 12 K for films synthesized on SrTiO_3 to ~ 20 K for films synthesized on LSAT⁵. This enhancement is attributed to optimization of the $R\text{NiO}_3$ parent perovskite, facilitated by the lower tensile strain (1.6% on LSAT versus 2.6% on SrTiO_3) in the as-grown state. Indeed, the lower tensile strain reduces the number of vertical rock-salt defects in the as-grown film which in turn enhances the T_c of the reduced, superconducting film.

As documented in our manuscript, there are subtle but critical differences between our materials and the infinite-layer compounds. First, we find a stronger propensity for the formation of vertical rock-salt defects in our films synthesized on substrates which impose tensile strain. This is perhaps unsurprising as our targeted composition $\text{Nd}_4\text{Ni}_2\text{O}_{10}$ is “A-site” rich compared to the perovskite phase NdNiO_3 . In contrast to the infinite layer compounds, therefore, this prevents us from stabilizing high-quality films under highly tensile conditions on SrTiO_3 . In addition, we find distinct strain-relieving defects in our reduced $\text{Nd}_4\text{Ni}_3\text{O}_8$ on compressively strained LaAlO_3 : rather than the a -oriented domains observed in the $R\text{NiO}_2$ compounds, we see a canting of the lattice. As has demonstrated in the infinite-layer compounds, however, careful consideration of the various strain-mediated defects and their

impact on further synthesis optimization is crucial for the advancement of the field.

REVIEWER #2 (REMARKS TO THE AUTHOR):

In the manuscript by Segedin et al., the authors reported the strain engineering of layered square-planar nickelate thin films. By synthesizing nickelate films on three different substrates, the authors explored the impact of strain on the crystalline quality of both the precursor Ruddlesden-Popper (RP) and reduced square-planar phases by XRD and STEM. Their results suggest that the crystalline quality of the RP films decreases as the tensile strain increases, resulting in best quality films on LaAlO₃ and worst quality films with disordered structure on SrTiO₃. As the reduction of the nickelate films experience a large expansion of the in-plane lattice constants, the films grown on NaGaO₃ has better lattice mismatch to accommodate such expansion and exhibit fair crystalline quality both before and after the reduction process. The authors also found that the rock-salt structures in RP films help stabilize the c-axis orientation of the reduced phase in the reduction process. The layered Nd_{n+1}Ni_nO_{2n+2} series are of great interests since they are predicted to exhibit more cuprate-like electronic structure and enhance T_c. However, the synthesis of such superconducting nickelate films has been an extremely challenging task. This work provides valuable information about the synthesis of high-quality films of this series. In the following, I have a few questions/concerns that I would like the authors to address.

1) It is clear that these three types of substrates used in this work can provide different strain states for the nickelate films, which is the reason that the authors attribute the observed phenomena to the strain effects. However, there are also some other distinct differences between these substrates. Especially, the SrTiO₃ has a non-polar structure, in contrast to the polar structure of other substrates. As the rock-salt layer has the tendency to increase the polar discontinuity, it could be harder to formed compared to the perovskite NdNiO₃. The rock-salt structure in RP nickelate films may be able to be stabilized on SrTiO₃ by properly engineering the interface structure, i.e. with a polar buffer layer. Nonetheless, it is important to rule out the polar discontinuity effect in order to make solid discussions on the strain effects.

Our response: We thank the reviewer for bringing up this important point. They are correct that the SrTiO₃ substrate and the nickelate film have different layered charge alterations, which would form a polar discontinuity at an abrupt, ideal atomic interface between the two. Indeed, the importance of this polar interface was also raised in the context

of superconducting infinite-layer films on SrTiO₃ as the possible source of a superconducting two-dimensional electron gas (2DEG) like the one that can be stabilized between SrTiO₃ and LaAlO₃. A detailed look at the real atomic interface which forms between SrTiO₃ and nickelate films – namely, a single-layer cation reconstruction and its impact on the electronic states across the interface – can be found in the preprint Goodge, et al. *arXiv:2201.03613* (2022). In short, atomic-scale chemical and lattice reconstruction at the interface between SrTiO₃ and NdNiO₂ fully alleviates the polar discontinuity between substrate and infinite-layer films.

To address the Reviewer’s question, we have performed additional STEM-EELS measurements of the film-substrate interfaces in each of the three systems we present in this work, which are now included in the Supplemental Material. Atomic-resolution elemental mapping of the interface between SrTiO₃ substrate and the as-grown Nd₄Ni₃O₁₀ film shows the formation of a similar intermediate layer as that observed in the infinite-layer nickelates, suggesting that the polar discontinuity is likely fully or mostly alleviated within ~ 1 unit cell. We further point out that the Ruddlesden-Popper fault mapping shown in Figure 5 of the Main Text in fact reveals an increase in the total density of rock salt planes (both horizontal and vertical) in the film grown on SrTiO₃. Furthermore, if the film were trending towards a perovskite structure as the Reviewer suggests, we would expect the HAADF-STEM images to show much clearer atomic column contrast between the *A*- and *B*- sites, rather than the largely reduced contrast that we observe in Figure 2.

In response to this question, we added “Supplementary Note 3: Atomic structure of the film-substrate interface.” The following figures are now included in the Supplemental Material:

We have also added the following text to the Methods:

Electron energy loss spectroscopy (EELS) measurements were carried out on the same instrument described above equipped with a Gatan Continuum spectrometer and camera. Spectrum images of the films on LaAlO₃ and NdGaO₃ were acquired with a spectrometer dispersion of 0.3 eV per channel. Spectrum images of the film on SrTiO₃ were acquired operating in DualEELS mode with a spectrometer dispersion of 0.15 eV per channel. The accelerating voltage was 300

FIG. R9: Simultaneously acquired annular dark-field (Sim. ADF) STEM image and atomic-resolution electron energy loss spectroscopy (EELS) elemental maps of the interface between the $\text{Nd}_4\text{Ni}_3\text{O}_{10}$ film and SrTiO_3 substrate shows a similar single unit cell intermediate layer of $\sim\text{Nd}(\text{Ti},\text{Ni})\text{O}_3$ as that observed and characterized for infinite-layer films on SrTiO_3 ⁶. Spectroscopic studies and theoretical calculations of the intermediate layer in perovskite/infinite-layer thin films show that it alleviates the strong polar discontinuity which would otherwise form at an abrupt interface between the charge neutral planes of SrTiO_3 and the charged planes in the nickelate film. The atomic contrast of Nd and Ni is washed out in most of the film due to the significant concentration of Ruddlesden-Popper faults. Summed concentration profiles for the Ti- $L_{2,3}$, Nd- $M_{4,5}$, and Ni- $L_{2,3}$ maps are also shown, along with a false color overlay of the Ti- $L_{2,3}$ (yellow) and Nd- $M_{4,5}$ (cyan) maps. Yellow and cyan arrows and dashed lines mark the atomic planes of the intermediate layer. The apparent skew of the ADF-STEM image and EELS maps is from sample drift during the EELS acquisition.

(120) kV for measurements of the films on SrTiO_3 and NdGaO_3 (LaAlO_3). Due to their overlapping EELS edges, two-dimensional concentration maps of the La- $M_{4,5}$ and Ni- $L_{2,3}$ edges are determined by non-negative least squares (NNLS) fit to the weighted sum of reference components for each edge taken from substrate (La- $M_{4,5}$) and film (Ni- $L_{2,3}$) regions (Supplemental Figure R11).

FIG. R10: Simultaneously acquired annular dark-field (Sim. ADF) STEM image and atomic-resolution electron energy loss spectroscopy (EELS) elemental maps of the interface between the $\text{Nd}_4\text{Ni}_3\text{O}_{10}$ film and NdGaO_3 substrate. The formal planar charge alterations of NdGaO_3 are ± 1 , close to the formal planar charges in the nickelate film. Summed concentration profiles for the Nd-M_{4,5} and Ni-L_{2,3} maps are also shown, along with a false color overlay of the Nd-M_{4,5} (cyan) and Ni-L_{2,3} (red) maps. The apparent skew of the ADF-STEM image and EELS maps is from sample drift during the EELS acquisition.

FIG. R11: Simultaneously acquired annular dark-field (Sim. ADF) STEM image and atomic-resolution electron energy loss spectroscopy (EELS) elemental maps of the interface between the $\text{Nd}_4\text{Ni}_3\text{O}_8$ film and LaAlO_3 substrate. The formal planar charge alterations of LaAlO_3 are ± 1 , close to the formal planar charges in the nickelate film. The half-unit of $\text{Nd}_4\text{Ni}_3\text{O}_8$ (i.e., single NiO_2 plane below the first fluorite layer) is due to a different shuttering sequence used in the growth of this film as compared to the full $\text{Nd}_4\text{Ni}_3\text{O}_{10}$ unit (three NiO_2 planes before the first rock salt plane) observed in the film grown on NdGaO_3 shown in Figure R10. Due to their overlapping EELS edges, two-dimensional concentration maps of the $\text{La-M}_{4,5}$ and $\text{Ni-L}_{2,3}$ edges are determined by non-negative least squares (NNLS) fit to the weighted sum of reference components for each edge taken from substrate ($\text{La-M}_{4,5}$) and film ($\text{Ni-L}_{2,3}$) regions marked by the yellow and red dashed boxes on the simultaneous ADF-STEM image. Summed concentration profiles for the $\text{La-M}_{4,5}$, $\text{Nd-M}_{4,5}$, and $\text{Ni-L}_{2,3}$ maps are also shown, along with a false color overlay of the $\text{La-M}_{4,5}$ (yellow), $\text{Nd-M}_{4,5}$ (cyan), and $\text{Ni-L}_{2,3}$ (red) maps. The reference components for the $\text{La-M}_{4,5}$ and $\text{Ni-L}_{2,3}$ concentration mapping are shown at the far right.

2) *In the manuscript, the author mainly discussed the strain effects based on the experimental observations. It would be helpful if the authors can provide theoretical simulations to support their arguments.*

Our response: We thank the Reviewer for the opportunity to improve our manuscript. We are pleased to add density-functional theory (DFT) calculations to explore the effects of in-plane strain on the basic features of the electronic structure of the layered nickelates. Our results show that (1) the charge-transfer energy (a measure of the p - d splitting) increases (decreases) with compressive (tensile) strain, (2) the rare-earth density-of-states at the Fermi level increases (decreases) with compressive (tensile) strain, and (3) the Ni- $3d_{x^2-y^2}$ bandwidth increases (decreases) with compressive (tensile) strain. Both the charge transfer energy and Ni $3d_{x^2-y^2}$ bandwidth are deemed important for cuprate superconductivity, which further motivates our experimental study of strain effects in the layered nickelates.

We have added the following to the main text:

Finally, our density-functional theory (DFT) calculations reveal that in-plane strain alters aspects of the $R_{n+1}Ni_nO_{2n+2}$ electronic structure relevant to superconductivity. Our study thus demonstrates a pathway to the synthesis of a superconducting compound and sets limits on the ability to strain-engineer $R_{n+1}Ni_nO_{2n+2}$ thin films via epitaxy.

Density-Functional Theory Calculations

We perform DFT calculations to investigate how strain tunes the electronic structure of $Nd_4Ni_3O_{10}$ and $Nd_6Ni_5O_{16}$. As detailed in Supplementary Note 5 and summarized schematically in Fig. R14, our calculations reveal the following effects:

1. The charge-transfer energy (Δ , a measure of the p - d splitting) increases (decreases) with compressive (tensile) strain.

2. The rare-earth density-of-states at the Fermi level increases (decreases) with compressive (tensile) strain.
3. The Ni- $d_{x^2-y^2}$ bandwidth increases (decreases) with compressive (tensile) strain.

The role of the charge-transfer energy Δ and rare-earth $5d$ ‘spectator’ bands in nickelate superconductivity has been extensively debated⁷⁻¹⁰. In the cuprates, the superexchange interaction $J \sim t^4/\Delta^3$ and single-band fermiology are widely deemed essential for superconductivity^{11,12}. Thus strain-engineering $R_{n+1}Ni_nO_{2n+2}$ compounds provides an avenue to explore the role of the charge-transfer energy and multi- or single-band fermiology in nickelate and cuprate superconductivity. Next we discuss the synthesis of these compounds under a variety of strain states.

...

Given the high density of extended defects observed in $Nd_4Ni_3O_{10}$ / $SrTiO_3$, we disqualify this system from consideration for topotactic reduction and instead turn our attention to substrates which yield smaller lattice mismatch. Our DFT calculations however predict that, of the three strain states studied here, $Nd_4Ni_3O_8$ / $SrTiO_3$ yields a fermiology closest to that of the cuprates due to the decreased charge transfer energy and minimal neodymium DOS at ε_F (Supplemental Note 1). In an effort to decrease the density of extended defects, we next synthesize $Nd_4Ni_3O_{10}$ under more modest 0.8% tensile strain on $NdGaO_3$ (Fig. 1).

...

Next we reduce $Nd_4Ni_3O_{10}$ on $LaAlO_3$ to investigate how rock salt layering impacts the structural transformation to the square-planar phase. The reduction-induced increase in compressive strain of the Ruddlesden-Popper (0.9% to 3.2%) is comparable to that of the perovskite (0.5% to 3.3%). In Fig. 6(c), we present XRD scans of $Nd_4Ni_3O_{10}$ and $Nd_4Ni_3O_8$ films on $LaAlO_3$. The $Nd_4Ni_3O_{10}$ / $LaAlO_3$ film exhibits all expected $n = 3$ Ruddlesden-Popper reflections and, as shown in Fig. 6(e), exhibits a metal-to-insulator transition at ~ 150 K, con-

sistent with previous reports^{13,14}. ... We note that, of the three strain states studied here, $R_{n+1}\text{Ni}_n\text{O}_{2n+2}$ films under high compressive strain on LaAlO_3 are the furthest from cuprate-like due to the larger charge-transfer energy and higher neodymium DOS at ε_F (Supplemental Note 1). Thus if optimized to a metallic state, $R_{n+1}\text{Ni}_n\text{O}_{2n+2} / \text{LaAlO}_3$ films could be an interesting platform to study the role of the charge transfer energy and neodymium $5d$ states in nickelate superconductivity.

...

Through our systematic study across multiple substrates, we establish a method to synthesize layered square-planar nickelate thin films and set limits on the ability to strain-engineer these compounds. Furthermore, our DFT calculations demonstrate that in-plane strain significantly tunes the $R_{n+1}\text{Ni}_n\text{O}_{2n+2}$ fermiology; thus if optimized to a metallic state on LaAlO_3 and SrTiO_3 , these systems could be exciting platforms to study the role of the charge-transfer energy and rare-earth $5d$ states in nickelate superconductivity.

We have also updated the Supplemental Material with the following discussion:

We employ density functional theory (DFT)-based calculations to understand basic trends in the electronic structure of the trilayer ($\text{Nd}_4\text{Ni}_3\text{O}_8$) and five-layer ($\text{Nd}_6\text{Ni}_5\text{O}_{12}$) nickelates with in-plane strain. Our calculations were performed using the projector augmented (PAW) method as implemented in VASP¹⁵ with the Perdew-Burke-Ernzerhof (PBE) version of the exchange correlation functional¹⁶. Note that we put the Nd($4f$) states in the pseudopotential core. For both materials, we used a $12 \times 12 \times 12$ \mathbf{k} -grid for the Brillouin zone integration. To model the effects of strain, we change the in-plane lattice constant, then optimize the atomic positions and out-of-plane lattice constants keeping the in-plane lattice constants fixed.

In the context of the cuprate superconductors, the charge-transfer energy (Δ_{CT}) and bandwidth of the $d_{x^2-y^2}$ bands are relevant parameters in connection to T_c . Estimates for both of these quantities can be derived from our electronic structure calculations. Figure R12(a) highlights how the charge-transfer energy (a measure of the hybridization between the Ni($3d$) and O($2p$) states) is influenced with strain. We estimate the charge-transfer energy, defined as $\Delta_{\text{CT}} = E_d - E_p$ from the band centroids: $E_\alpha = \int dE g_\alpha(E)E / \int dE g_\alpha(E)$, where g_α is the partial density of states for orbital α and $\alpha \in [\text{Ni-}d_{x^2-y^2}, \text{O-}p_\sigma]$. We find that compressive ($\varepsilon < 0$) strain increases the charge-transfer energy (pushing the Ni($3d$) and O($2p$) states apart), while tensile strain ($\varepsilon > 0$) strain decreases the charge-transfer energy (this is trend is also captured in Fig. R13(c,d)).

The nearest-neighbor hopping t between the Ni- $d_{x^2-y^2}$ orbitals is an important parameter as it sets the energy scale for the low-energy physics of the material. We estimate t by fitting the DFT bands corresponding to the Ni- $d_{x^2-y^2}$ states to a tight-binding model on a square lattice,

$$\varepsilon(\mathbf{k}) = -\mu - 2t(\cos(k_x a) + \cos(k_y a)) - 4t'(\cos(k_x a)\cos(k_y a)) - 4t''(\cos(2k_x a) + \cos(2k_y a)), \quad (\text{R1})$$

FIG. R12: Basic features of the electronic structure upon strain for $\text{Nd}_4\text{Ni}_3\text{O}_8$ (blue) and $\text{Nd}_6\text{Ni}_5\text{O}_{12}$ (red): (a) charge-transfer energy (Δ_{CT}), (b) nearest-neighbor hopping (t), and (c) the neodymium density of states at the Fermi level (ϵ_F). The strain convention is the same one used in the main text: $\epsilon = (a_{\text{sub}} - a_{\text{bulk}})/a_{\text{bulk}}$. Note that -1.5% compressive strain corresponds to $\text{Nd}_4\text{Ni}_3\text{O}_{10}$ / NdGaO_3 .

where μ is the chemical potential, t is the nearest-neighbor, t' is the next-nearest-neighbor, and t'' is the third-nearest-neighbor hopping parameters. Figure R12(b) reveals the expected trend in the nearest-neighbor hopping. For in-plane compressive strain, the orbital overlap of Ni atoms is increased, which manifests in an overall increase in the $d_{x^2-y^2}$ bandwidth. Similarly, in-plane tensile strain decreases the bandwidth. Utilizing strain to control the overall bandwidth (and energy scale) has been suggested as a viable mechanism for enhancing T_c in the infinite-layer nickelates⁸.

Finally, Fig. R12(c) highlights the evolution of the density of states (DOS) corresponding to the neodymium states at the Fermi level (ϵ_F). We observe that the neodymium DOS at ϵ_F is finite for $\epsilon < -1.5\%$, and approximately zero for $\epsilon > -1.5\%$. Thus strain has an influence on the magnitude of the neodymium DOS at ϵ_F similar to chemical doping¹⁷.

Figure R13 displays a complete summary of the evolution of the band structure and density of states for the DFT calculations as function of strain from which the basic features that we describe above were derived. The overall trend in the charge-transfer energy can be visually seen from the evolution of the density of states in Fig. R13(c,d) and the decreasing bandwidth of the Ni- $d_{x^2-y^2}$ bands can be seen in Fig. R13(a,b). The trends in charge transfer energy and neodymium

DOS at ε_F are schematically summarized in Fig. R14.

FIG. R13: DFT electronic structure for Nd₄Ni₃O₈ and Nd₆Ni₅O₁₂ upon applied strain. (Top panels) Paramagnetic band structures as function of strain for Nd₄Ni₃O₈ (a) and Nd₆Ni₅O₁₂ (b). (Bottom panels) Atom-resolved density of states (DOS) as a function of strain for Nd₄Ni₃O₈ (c) and Nd₆Ni₅O₁₂ (d).

FIG. R14: Schematic cartoon highlighting the effect of compressive and tensile strain on the fermiology of $\text{Nd}_4\text{Ni}_3\text{O}_8$. We show the influence of strain on the charge transfer energy (Fig. R12(a)) and neodymium 5d density of states at ϵ_F (Fig. R12(c)). The effect of strain on the nickel 3d bandwidth is not shown (Fig. R12(b)), as it is a minor effect in comparison to the changes in charge transfer energy and neodymium DOS.

3) *It is there any reason why Nd₄Ni₃O₁₀/LAO show substantial better crystallinity than Nd₄Ni₃O₁₀/NGO as the absolute values of the lattice mismatch are nearly the same (1%) for Nd₄Ni₃O₁₀ grown on these two substrates. Could the authors comment on this point?*

Our response: We thank the author for raising this important point. Indeed, Nd₄Ni₃O₁₀ films on LaAlO₃ and NdGaO₃ exhibit approximately the same magnitude of epitaxial strain (0.9% compressive and 0.8% tensile, respectively). Crucially, however, the **sign** of the lattice mismatch determines what kind of defects occur in order to relieve the strain. For example, it was shown that compressive strain in LaNiO₃ is relieved by misfit dislocations while vertical rock salt faults relieve tensile strain¹⁸.

To further address this point, we have added the following discussion to the supplemental materials:

We estimate the relationship between the density of vertical rock salt faults density and effective lattice mismatch in nickelate thin films. We define the effective lattice mismatch as

$$\epsilon_{eff} = \frac{l_{sub} - l_{film}}{l_{film}}$$

where

$$l_{sub} = n_{sub} \cdot a_{sub}$$

$$l_{film} = n_{film} \cdot a_{film} + n_{RP} \cdot d_{RP}$$

As illustrated in Fig. S4(a), n_{sub} and n_{film} denote the number of substrate and film unit cells, respectively, along the in-plane direction across a given lateral span. The substrate and film lattice constants are given by a_{sub} and a_{film} , respectively. The monolayer spacing between neodymium planes within a rock

salt (RP) fault is denoted by d_{RP} , while n_{RP} represents the number of vertical RP faults in the same lateral span defined by n_{sub} and n_{film} . For a minimal fault density, the insertion of one vertical RP fault necessitates the formation of a second in order to maintain a coherent interface, as shown in Fig. S4(a). We thus set $n_{RP} = 2$. With the insertion of 2 vertical RP faults, two film monolayers (i.e. one unit cell) must be removed, giving $n_{film} = n_{sub} - 1$.

In Fig. S4(b), we plot the effective lattice mismatch

$$\epsilon_{eff}(n_{sub}) = \frac{(n_{sub} \cdot a_{sub}) - ((n_{sub} - 1) \cdot a_{film} + 2d_{RP})}{(n_{sub} - 1) \cdot a_{film} + 2d_{RP}}$$

for $\text{Nd}_4\text{Ni}_3\text{O}_{10}$ ($a_{film} = 3.826 \text{ \AA}$) and NdNiO_3 ($a_{film} = 3.807 \text{ \AA}$), with $d_{RP} = 2.76 \text{ \AA}$ ¹⁹ for the three substrates studied here: SrTiO_3 ($a_{sub} = 3.905 \text{ \AA}$), NdGaO_3 ($a_{sub} = 3.858 \text{ \AA}$), and LaAlO_3 ($a_{sub} = 3.790 \text{ \AA}$). This calculation demonstrates how the insertion of vertical rock salt faults in nickelate films grown under tensile strain can effectively decrease the lattice mismatch. In fact, the insertion of approximately 4 and 9 vertical rock salt faults per 100 unit cells can effectively eliminate the lattice mismatch in $\text{Nd}_4\text{Ni}_3\text{O}_{10}$ grown on NdGaO_3 and SrTiO_3 , respectively. For films grown under compressive strain, on the other hand, the insertion of vertical RP faults only further increases the magnitude of the lattice mismatch. Therefore, $\text{Nd}_4\text{Ni}_3\text{O}_{10}$ films grown on NdGaO_3 are more susceptible to the formation of tensile strain relieving RP faults than those grown on LaAlO_3 , despite the fact that the magnitude of the lattice mismatch for both is approximately equal. Our calculation qualitatively describes the strain-dependent density of vertical RP faults we observe in Fig. 5 of the main text.

We reference this discussion in the main text as follows:

Such an increase in rock salt fault density is expected as the vertical Ruddlesden-Popper layers relieve tensile strain^{18,20}, as discussed in Supplementary Note 1.

...

FIG. R15: (a) Schematic crystal structure of an epitaxial thin film with a rock salt Ruddlesden-Popper (RP) fault, highlighted by the yellow line. The specified values for a_{film} and d_{RP} correspond to $\text{Nd}_4\text{Ni}_3\text{O}_{10}$ ¹⁹. (b) The effective lattice mismatch calculated for $\text{Nd}_4\text{Ni}_3\text{O}_{10}$ and NdNiO_3 thin films on LaAlO_3 , NdGaO_3 , and SrTiO_3 with varying density of vertical RP faults, plotted as a function of (top x -axis) the average fault separation (n_{sub}) and (bottom x -axis) the RP fault density per 100 substrate unit cells ($100 \cdot (2/n_{sub})$).

Fig. 5 caption: ... See Supplemental Figure S4 for a discussion regarding the observed strain-dependent defect formation.

4) The XRD patterns indicate that the stoichiometry in present RP films grown on different substrates deviates from the ideal values (the Δc is not identical for two adjacent 00l diffraction peaks). Therefore, I am curious about what role does the stoichiometry play in terms of structural and transport properties, because it seems the metallic behavior is more robust for films grown on NGO than LAO, although the disorder level is very similar for these two cases.

Our response: We thank the reviewer for their insightful comments. We first note that the vertical solid and dotted lines in our x-ray figures do not denote 'ideal' peak locations, but rather peak locations of the bulk compound (as stated in the figure captions). Thus for our epitaxial thin films, the deviation from the bulk 00l peak locations (i.e. deviation from bulk c -axis lattice constant) corresponds to a contraction (expansion) of the c -axis lattice constant under tensile (compressive) strain. We have added the following figure to the Supplemental Materials to illustrate this point:

FIG. R16: (a) XRD scans of $\text{Nd}_4\text{Ni}_3\text{O}_{10}$ films grown on NdGaO_3 ($\epsilon = +0.8\%$) and LaAlO_3 ($\epsilon = -0.9\%$). (b) c -axis lattice constants calculated for the films shown in (a). The trend (b) confirms the expected expansion (compression) of the c -axis lattice constant with compressive (tensile) strain, and demonstrates that the films are epitaxially strained to the substrate. The lattice constants are calculated via Nelson-Riley fits of the XRD peaks.

We also thank the reviewer for bringing up stoichiometry, a factor that is crucial to understand the nature of metallicity in reduced $\text{Nd}_4\text{Ni}_3\text{O}_8$ compounds. To address this point, we have synthesized a series of five $\text{Nd}_4\text{Ni}_3\text{O}_{10}$ / NdGaO_3 with systematically varying neodymium content. With this sample set, we have performed extensive chemical reductions

as well as electrical transport, structural, and x-ray absorption spectroscopy measurements to investigate the role of cation stoichiometry in the metallicity of $\text{Nd}_4\text{Ni}_3\text{O}_8$ / NdGaO_3 films. In summary, we have demonstrated that the metallicity of reduced $\text{Nd}_4\text{Ni}_3\text{O}_8$ films is exquisitely sensitive to the cation stoichiometry: samples that are either 3% neodymium rich or deficient are consistently insulating upon reduction, despite being structurally comparably to the optimal metallic sample.

We have added expanded discussions on the effect of cation stoichiometry to the Main Text and as Supplemental Note 8, as included above / below (pages 36- 42 of this response letter).

REVIEWER #3 (REMARKS TO THE AUTHOR):

This is a beautiful study of the structural and transport properties of nickelate RP films and their reduced variants. Currently, this group is the only one in the world who has been able to make and study such reduced RP films. As such, I strongly support the publication of this paper in Nature Communications.

What the paper lacks, though, is a comparison to previous studies of reduced bulk materials. The first paper I would like to point out is Retoux et al (J. Solid State Chem. 140, 307 (1998)) who studied defects in bulk Nd₄38. They primarily found stacking faults with reduced n ($n=1$ and 2), but occasionally found ones of higher n (like $n=6$). A brief comparison to this paper would be in order.

Our Response: We thank the Reviewer for their careful consideration of our manuscript. We agree that a comparison to the defects observed in bulk Nd₄Ni₃O₈ would strengthen our paper. As the Reviewer points out, the primary defects observed in bulk Nd₄Ni₃O₈ are intergrowths of $n = 1$ and $n = 2$ regions with fewer $n = 6$ regions²¹. These defects are also evident in the oxidized La₄Ni₃O₁₀ and La₃Ni₂O₇ compounds^{22,23}. It is important to note that the cation disorder in the reduced Nd₄Ni₃O₈ compound originates from the oxidized Nd₄Ni₃O₁₀ precursor due to the minimal cation mobility at typical reduction temperatures ($\sim 300^\circ\text{C}$). Our films exhibit a small density of such deviations (Fig. S7 in¹⁴) due to the MBE shuttering sequence that encourages the formation of the targeted Ruddlesden-Popper order. In other, strain-free RPs²⁴ there are stacking faults that form to accommodate off-composition.

We have made the following additions in the main text to address this point:

MAADF-STEM images of this film in Fig. 4(c) show nearly uniform adherence to horizontal layering with very few vertical rock salt faults. Atomic-resolution HAADF-STEM imaging of the film (Fig. 4(d)) further confirms the high degree of crystalline order, with only few defects such as deviations from the $n = 3$ Ruddlesden-Popper structure, quantified in Fig. S7 in Ref.¹⁴. While such deviations from the targeted $n = 3$ Ruddlesden-Popper structure are observed in La₄Ni₃O₁₀^{22,23} and reduced Nd₄Ni₃O₈²¹ bulk crystals, our films exhibit a small density of such deviations due to the MBE shuttering sequence that encourages the formation of the targeted Ruddlesden-Popper order.

...

With the challenges in stabilizing the parent compounds, $\text{Nd}_{n+1}\text{Ni}_n\text{O}_{3n+1}$, in mind, we next consider their oxygen deintercalation. We note that the cation disorder observed in the parent compounds is preserved through reduction due to the minimal cation mobility at typical reduction temperatures ($\sim 300^\circ\text{C}$).

Second, I looked through the literature and found all the papers I could where transport was measured on bulk $\text{Nd}_4\text{Ni}_3\text{O}_8$. This involved four published papers along with a slide I had from a conference proceedings. Most of these found insulating behavior like the authors find for $\text{Nd}_4\text{Ni}_3\text{O}_8$ on LAO. But one of these papers (Miyatake et al.) found results analogous to what the authors find for $\text{Nd}_4\text{Ni}_3\text{O}_8$ on NGO. These materials were processed with sulfur, which they claim acts as an oxygen getter. This implies that this difference might be due to stoichiometry and thus the Ni valence. It could well be that the density wave state in the $\text{Nd}_4\text{Ni}_3\text{O}_8$ materials that is associated with insulating behavior only exists in a narrow doping range, similar to what is seen for some of the charge ordered states in layered manganites.

In that context, the authors early on in the paper discuss Nd enriched samples and associated insertion of extra NdO layers, but this argumentation did not seem to be extended to the reduced materials. Imagine that one inserts x layers per formula unit of NdO. Then the resulting nickel valence will be reduced by $x/3$. That is, one would effectively be electron doping relative to $1/3$. Perhaps such a reduced Ni valence relative to $1/3$ is associated with metallic behavior? This is worth some thought.

Our Response: The Reviewer brings up an important point regarding the relation between stoichiometry and metallicity in $\text{Nd}_4\text{Ni}_3\text{O}_8$ compounds. We first address the question of **oxygen** content. As the Reviewer notes, Miyatake *et al.* were able to stabilize metallic $\text{Nd}_{3.5}\text{Sm}_{0.5}\text{Ni}_3\text{O}_8$ compounds for the first time by using sulfur as an oxygen getter^{25,26}. The authors in these papers stress that because $\text{Nd}_4\text{Ni}_3\text{O}_8$ is a T' structure like the T' cuprates, it may be essential to fully remove apical oxygens from the system in order to stabilize metallicity (and superconductivity), as was shown for the T' cuprates²⁷. While oxygen content is certainly important to consider, we are confident that we are fully reducing our samples using the well-established CaH_2 reduction technique as we have been able to reproducibly achieve metallic and even superconducting samples²⁸ (see response to reviewer #1 for detailed reduction trials).

We have incorporated the above discussion in the following paragraph in the main text:

Additionally, the dramatic expansion (contraction) of the in-plane (out-of-plane) lattice parameter through reduction further complicates the stabilization of high quality square-planar nickelates. For example, reduced $\text{La}_{1-x}\text{Ca}_x\text{NiO}_2$ single crystals exhibit three orthogonally-oriented domains of the infinite-layer phase separated by micro-cracks²⁹. While not superconducting, $\text{La}_{1-x}\text{Ca}_x\text{NiO}_2$ and $\text{Pr}_4\text{Ni}_3\text{O}_8$ ²⁹ single crystals are metallic, unlike most reduced infinite-layer^{30–33} and layered nickelate powders to date^{34–40}. A notable exception is the case of metallic $(\text{Pr,Nd})_4\text{Ni}_3\text{O}_8$ pellets which were reduced using a sulfur getter to further promote the removal of apical oxygens^{25,26}. Thus metallicity in the T' nickelates may be exquisitely sensitive to the oxygen content like in the T' cuprates²⁷. Additional proposals for the the lack of superconductivity in bulk reduced nickelates include nickel deficiency and the nucleation of ferromagnetic nickel grains that suppress superconductivity⁴¹.

Next we consider the role of **cation** stoichiometry (Nd / Ni ratio). As the Reviewer notes, the insertion of additional neodymium may effectively electron dope the system: $\text{Nd}_{4+x}\text{Ni}_3\text{O}_8$ would yield a $3d^{8.67+x}$ electron count. To address this comment, we have synthesized a series of five $\text{Nd}_4\text{Ni}_3\text{O}_{10}$ / NdGaO_3 with systematically varying neodymium content. With this sample set, we have performed extensive chemical reductions as well as electrical transport, structural, and x-ray absorption spectroscopy measurements to investigate the role of cation stoichiometry in the metallicity of $\text{Nd}_4\text{Ni}_3\text{O}_8$ / NdGaO_3 films. In summary, we have demonstrated that the metallicity of reduced $\text{Nd}_4\text{Ni}_3\text{O}_8$ films is exquisitely sensitive to the cation stoichiometry: samples that are either 3% neodymium rich or deficient are consistently insulating upon reduction, despite being structurally comparably to the optimal metallic sample.

We have added expanded discussions on the effect of cation stoichiometry to the Main Text and as Supplemental Note 8, as included above / below (pages 36- 42 of this response letter).

A. Cation stoichiometry

While reduced $\text{Nd}_{n+1}\text{Ni}_n\text{O}_{2n+2}$ films on NdGaO_3 are metallic ($n = 3$) and superconducting ($n = 5$)²⁸, the metallicity of these films is highly sensitive to as-synthesized structural differences and reduction conditions (Fig. R6 and Fig. S2 in Ref.²⁸). In the infinite-layer nickelates, cation stoichiometry has been shown to strongly influence metallicity and superconductivity^{1,3}. We thus investigate the role of cation stoichiometry in the metallicity of reduced layered nickelates. In Fig. R17(a) we present XRD scans of five $\text{Nd}_4\text{Ni}_3\text{O}_{10}$ films with systematically varying neodymium content (see Supplemental Note 8 for more details on the MBE synthesis of these films). Electrical transport measurements shown in Fig. R17(b-c) demonstrate that the resistivity of the as-synthesized films is relatively insensitive to the cation stoichiometry, except for the the 6% neodymium-rich sample which exhibits higher resistivity than the rest of the samples in the series. More details on the influence of cation stoichiometry on the transport and electronic properties of these $\text{Nd}_4\text{Ni}_3\text{O}_{10}$ films can be found in Figs. R19 and Fig. R20.

Next we reduce the sample series shown in Fig. R17(a). The resulting XRD scans are displayed in Fig. R17(d). While the three samples within 3% of optimal stoichiometry exhibit all $\text{Nd}_4\text{Ni}_3\text{O}_8$ $00l$ peaks, the two samples that deviate from optimal stoichiometry by 6% exhibit very few of those peaks. While the structural differences between the samples within 3% of optimal stoichiometry are minor, the transport properties shown in Fig. R17(e-f) reveal striking differences. Remarkably, only the sample with optimal neodymium content exhibits metallic transport upon reduction (see Figs. R3 and S17 for more trials). Even though the three samples within 3% of optimal stoichiometry are structurally comparable by XRD, the resistivity of the optimal sample differs from the rest by more than an order of magnitude (Fig. R17(f)). Therefore, the metallicity of $\text{Nd}_4\text{Ni}_3\text{O}_8 / \text{NdGaO}_3$ is exquisitely sensitive to the cation stoichiometry.

FIG. R17: Structural and electrical transport characterization of (a-c) $\text{Nd}_4\text{Ni}_3\text{O}_{10}$ and (d-f) $\text{Nd}_4\text{Ni}_3\text{O}_8$ films on NdGaO_3 (110) with varying cation neodymium content. All films are reduced for 3 hours at 290°C . (a) XRD scans of $\text{Nd}_4\text{Ni}_3\text{O}_{10} / \text{NdGaO}_3$ films with varying neodymium content (controlled by shutter times in the MBE) within 6% of the optimal value. (b) Resistivity versus temperature and (c) room temperature resistivity measurements of the $\text{Nd}_4\text{Ni}_3\text{O}_{10}$ films in (a). (d) XRD scans of the reduced $\text{Nd}_4\text{Ni}_3\text{O}_8$ films from (a). The peak at $\sim 37^\circ$ in (a) and (b) is the gold 111 peak from electrical contacts. (e) Resistivity versus temperature and (f) room temperature resistivity measurements of the reduced $\text{Nd}_4\text{Ni}_3\text{O}_8$ films in (d). The filled markers in (f) correspond resistivity data in (e). The empty markers in (f) denote resistivity measurements via multimeter of other pieces of the same sample, all reduced at 3 hours at 290°C . More chemical reduction trials of these samples can be found in Figs. R3 and S17. The resistivity versus temperature measurements for the insulating samples were stopped once reaching compliance voltage limit in PPMS.

SUPPLEMENTAL NOTE 8: CATION STOICHIOMETRY

To investigate the role of cation stoichiometry (Nd/Ni ratio), we synthesized a series of five $\text{Nd}_4\text{Ni}_3\text{O}_{10}$ / NdGaO_3 films with varying neodymium content, which we will refer to as the 'stoichiometry series'. We synthesized the series within twenty four hours to ensure chamber stability, which we further verify by re-synthesizing an duplicate sample reproducing the first film grown in the series to check for consistency. We vary the neodymium content by altering the neodymium monolayer shutter time between samples while keeping the nickel shutter time and all other growth parameters constant. As shown in Fig. R18, we vary the neodymium content by 3% between samples. Due to the difficulty in precisely measuring the neodymium content in neodymium nickelate films on NdGaO_3 by RBS, we do not assign compositions to the samples and instead refer to the percentage of neodymium relative to the optimal sample, which exhibits the highest intensity XRD peaks (Fig. R18) and lowest resistivity (Fig. R17).

In Fig. R18 we present the structural characterization of the $\text{Nd}_4\text{Ni}_3\text{O}_{10}$ / NdGaO_3 stoichiometry series. The optimal sample in Fig. R18(a) exhibits the highest intensity superlattice peaks of all samples in the series and is the only sample with a clear 006 peak. The most off-stoichiometric samples (-6% and +6% neodymium) exhibit much lower intensity XRD peaks; notably, the neodymium-deficient (-6%) sample possesses a considerably inferior structure to the neodymium-rich (+6%) film. The systematic left-ward shift in the 0016 peak location with increasing neodymium content highlighted in Fig. R18(b) is indicated of an expansion in c -axis lattice constant, shown in Fig. R18(c). The contracted c -axis lattice constants with respect to the bulk $\text{Nd}_4\text{Ni}_3\text{O}_{10}$ value is a consequence of the tensile strain (+0.8%). The observed expansion in c -axis lattice constant with neodymium-richness is consistent with neodymium-rich NdNiO_3 films^{3,42}.

Fig. R18 displays the structural characterization of the reduced $\text{Nd}_4\text{Ni}_3\text{O}_8$ stoichiometry series. The three samples within 3% of the optimal neodymium content in Fig. R18(d) exhibit all expected XRD peaks while the off-stoichiometric $\pm 6\%$ show very few peaks. Fig. R18(e) show hints of a systematic trend in 008 peak position with neodymium content, but the large error in the c -axis lattice parameter estimates shown in Fig. R18(f) preclude the

determination of a trend similar to the one evident in the as-grown samples in Fig. R18(c). The expanded c -axis lattice constants are consistent with the nominal compressive strain in the reduced state.

FIG. R18: Structural characterization of (a-c) $\text{Nd}_4\text{Ni}_3\text{O}_{10}$ and (d-f) $\text{Nd}_4\text{Ni}_3\text{O}_8$ films on NdGaO_3 (110) with varying neodymium content. All films are reduced for 3 hours at 290°C . (a) XRD scans of $\text{Nd}_4\text{Ni}_3\text{O}_{10} / \text{NdGaO}_3$ films with varying neodymium content (controlled by shutter times in the MBE) within 6% of the optimal value. (b) Zoom-in of the 0016 peaks. (c) c -axis lattice constants calculated by Nelson-Riley fits of the superlattice peaks in (a). (d) XRD scans of $\text{Nd}_4\text{Ni}_3\text{O}_8 / \text{NdGaO}_3$ films reduced from those shown in (a). (e) Zoom-in of the 008 peaks. (f) c -axis lattice constants calculated by Nelson-Riley fits of the superlattice peaks in (d).

Next we investigate how off-stoichiometry influences the electronic properties of $\text{Nd}_4\text{Ni}_3\text{O}_{10}$. Via formal electron counting, we can estimate that $\text{Nd}_{4+x}\text{Ni}_3\text{O}_{10}$ will possess a $+2.67-x$ nickel valence and $3d^{7.33+x}$ electron count. Therefore, excess neodymium may effectively electron dope the system. Measurements of the hall coefficient in Fig. R19(a) suggest a trend in carrier density with neodymium content at low temperature, shown in Fig. R19(b). Taking the hall coefficients measured at 10K, the carrier density decreases with increasing neodymium content (Fig. R19(c)).

FIG. R19: (a) Hall coefficient versus temperature measurements of $\text{Nd}_4\text{Ni}_3\text{O}_{10} / \text{NdGaO}_3$ (110) films with varying neodymium content. (b) Carrier density versus temperature and (c) carrier density at 10K versus neodymium content.

We can also verify changes to the nickel filling in the $\text{Nd}_4\text{Ni}_3\text{O}_{10}$ using x-ray absorption spectroscopy (XAS). In Fig. R20(a), we note the presence of two multiplet peaks at the nickel L_3 edge, labelled A and B. As we increase the neodymium content, we see a consistent diminishing of the peak B intensity relative to the peak A intensity. The relative increase of the A/B intensity ratio has been associated with a shift in nickel valence (filling) from $3+$ (d^7) to $2+$ (d^8) irrespective of the method of charge transfer, including: oxygen vacancy formation^{43–45}, modulation of Ruddlesden-Popper order¹⁴, and interfacial doping⁴⁶. The overall width of the nickel L_3 edge also decreases with increasing neodymium, further reflecting a decrease in intensity of the B peak, which is the broader of the two multiplet features. Here then, the XAS clearly suggest that the formal nickel valence (filling) decreases (increases) with increasing neodymium content.

However, after reducing all five samples to the $\text{Nd}_4\text{Ni}_3\text{O}_8$ phase, we see no clear trends across the nickel L_3 edge (Fig. R20(b)). Assuming a perfectly oxygen-stoichiometric phase, if neodymium off-stoichiometry could effectively dope the nickel site, we would expect the relative broadening of the nickel L_3 edge to increase with increasing neodymium^{47,48}. We do not see this behavior. This indicates that the reduction process introduces an inherent randomness that potentially targets the oxygen stoichiometry, which may wash out obvious trends from cationic stoichiometry in spectroscopic probes. Nonetheless, the resistivity

FIG. R20: Nickel L_3 edge x-ray absorption spectra of the (a) as-synthesized $\text{Nd}_4\text{Ni}_3\text{O}_{10}$ and (b) reduced $\text{Nd}_4\text{Ni}_3\text{O}_8$ phases for a series of films with varying neodymium content. The inset in (a) is a zoom-in of the ‘B’ multiplet peak to highlight the decrease in B peak intensity with increasing neodymium content.

variation across the $\text{Nd}_4\text{Ni}_3\text{O}_8$ (Fig. R17(f)) still suggests that the stoichiometric composition of the as-synthesized $\text{Nd}_4\text{Ni}_3\text{O}_{10}$ phase is paramount in achieving metallic behavior in the reduced $\text{Nd}_4\text{Ni}_3\text{O}_8$ phase.

We note that in the octahedrally-coordinated nickelates, a change in nickel valence from 3+ to 2+ should also be accompanied by a very gradual shifting of peak A to lower energy⁴³. However, due to the incident energy variation at the endstation, which can be up to 0.4 eV, we cannot resolve this shift confidently. As a result, we do not attempt to extract this potential trend, and have artificially shifted the spectra so that the onset of the nickel L_3 edges align to the average onset of the five spectra, for both the $\text{Nd}_4\text{Ni}_3\text{O}_{10}$ and $\text{Nd}_4\text{Ni}_3\text{O}_8$ cases. This enables a clearer visualization of relative peak intensities and widths. We have also scaled the data to the intensity at the pre-edge, and performed a min-max normalization over the entire nickel L_3 edge.

¹ K. Lee, B. H. Goodge, D. Li, M. Osada, B. Y. Wang, Y. Cui, L. F. Kourkoutis, and H. Y. Hwang, *APL Materials* **8**, 041107 (2020).

² S. Zeng, C. S. Tang, X. Yin, C. Li, M. Li, Z. Huang, J. Hu, W. Liu, G. J. Omar, H. Jani, Z. S. Lim, K. Han, D. Wan, P. Yang, S. J. Pennycook, A. T. S. Wee, and A. Ariando, *Physical*

- Review Letters* **125**, 147003 (2020).
- ³ Y. Li, W. Sun, J. Yang, X. Cai, W. Guo, Z. Gu, Y. Zhu, and Y. Nie, *Frontiers in Physics* **9**, 719534 (2021).
 - ⁴ S. W. Zeng, X. M. Yin, C. J. Li, L. E. Chow, C. S. Tang, K. Han, Z. Huang, Y. Cao, D. Y. Wan, Z. T. Zhang, Z. S. Lim, C. Z. Diao, P. Yang, A. T. S. Wee, S. J. Pennycook, and A. Ariando, *Nature Communications* **13**, 743 (2022).
 - ⁵ K. Lee, B. Y. Wang, M. Osada, B. H. Goodge, T. C. Wang, Y. Lee, S. Harvey, W. J. Kim, Y. Yu, C. Murthy, S. Raghu, L. F. Kourkoutis, and H. Y. Hwang, (2022), 10.48550/arxiv.2203.02580.
 - ⁶ B. H. Goodge, B. Geisler, K. Lee, M. Osada, B. Y. Wang, D. Li, H. Y. Hwang, R. Pentcheva, and L. F. Kourkoutis, arXiv preprint arXiv:2201.03613 (2022).
 - ⁷ A. S. Botana and M. R. Norman, *Phys. Rev. X* **10**, 011024 (2020).
 - ⁸ M. Kitatani, L. Si, O. Janson, R. Arita, Z. Zhong, and K. Held, *npj Quantum Materials* **5**, 59 (2020).
 - ⁹ J. Karp, A. Hampel, and A. J. Millis, *Phys. Rev. B* **105**, 205131 (2022).
 - ¹⁰ Z. Li and S. G. Louie, arXiv:2210.12819 (2022).
 - ¹¹ D. J. Scalapino, *Rev. Mod. Phys.* **84**, 1383 (2012).
 - ¹² S. M. O'Mahony, W. Ren, W. Chen, Y. X. Chong, X. Liu, H. Eisaki, S. Uchida, M. H. Hamidian, and J. C. S. Davis, *Proceedings of the National Academy of Sciences* **119**, e2207449119 (2022), 2108.03655.
 - ¹³ W. Sun, Y. Li, X. Cai, J. Yang, W. Guo, Z. Gu, Y. Zhu, and Y. Nie, *Physical Review B* **104**, 184518 (2021).
 - ¹⁴ G. A. Pan, Q. Song, D. F. Segedin, M.-C. Jung, H. El-Sherif, E. E. Fleck, B. H. Goodge, S. Doyle, D. C. Carrizales, A. T. N'Diaye, P. Shafer, H. Paik, L. F. Kourkoutis, I. E. Baggari, A. S. Botana, C. M. Brooks, and J. A. Mundy, *Physical Review Materials* **6**, 055003 (2022).
 - ¹⁵ G. Kresse and J. Furthmüller, *Phys. Rev. B* **54**, 11169 (1996).
 - ¹⁶ J. P. Perdew, K. Burke, and M. Ernzerhof, *Phys. Rev. Lett.* **77**, 3865 (1996).
 - ¹⁷ H. LaBollita, M.-C. Jung, and A. S. Botana, (2022), 10.48550/arxiv.2206.01701.
 - ¹⁸ J. Bak, H. B. Bae, C. Oh, J. Son, and S.-Y. Chung, *The Journal of Physical Chemistry Letters* **11**, 7253 (2020).
 - ¹⁹ A. Olafsen, H. Fjellvåg, and B. C. Hauback, *Journal of Solid State Chemistry* **151**, 46 (2000).
 - ²⁰ C. Yang, Y. Wang, D. Putzky, W. Sigle, H. Wang, R. A. Ortiz, G. Logvenov, E. Benckiser,

- B. Keimer, and P. A. van Aken, *Symmetry* **14**, 464 (2022).
- ²¹ R. Retoux, J. Rodriguez-Carvajal, and P. Lacorre, *Journal of Solid State Chemistry* **140**, 307 (1998).
- ²² R. Ram, L. Ganapathi, P. Ganguly, and C. Rao, *Journal of Solid State Chemistry* **63**, 139 (1986).
- ²³ J. Drennan, C. Tavares, and B. Steele, *Materials Research Bulletin* **17**, 621 (1982).
- ²⁴ N. M. Dawley, B. H. Goodge, W. Egger, M. R. Barone, L. F. Kourkoutis, D. J. Keeble, and D. G. Schlom, *Applied Physics Letters* **117**, 062901 (2020).
- ²⁵ A. Nakata, S. Yano, H. Yamamoto, S. Sakura, Y. Kimishima, and M. Uehara, *Advances in Condensed Matter Physics* **2016**, 1 (2016).
- ²⁶ T. Miyatake, S. Shibutani, K. Hamada, J. Gouchi, Y. Uwatoko, K. Wakiya, I. Umehara, and M. Uehara, *Proceedings of the International Conference on Strongly Correlated Electron Systems (SCES2019)* (2020), 10.7566/jpscp.30.011061.
- ²⁷ M. Brinkmann, T. Rex, M. Stief, H. Bach, and K. Westerholt, *Physica C: Superconductivity* **269**, 76 (1996).
- ²⁸ G. A. Pan, D. F. Segedin, H. LaBollita, Q. Song, E. M. Nica, B. H. Goodge, A. T. Pierce, S. Doyle, S. Novakov, D. C. Carrizales, A. T. N'Diaye, P. Shafer, H. Paik, J. T. Heron, J. A. Mason, A. Yacoby, L. F. Kourkoutis, O. Erten, C. M. Brooks, A. S. Botana, and J. A. Mundy, *Nature Materials* **21**, 160 (2021).
- ²⁹ P. Puphal, Y.-M. Wu, K. Fürsich, H. Lee, M. Pakdaman, J. A. N. Bruin, J. Nuss, Y. E. Suyolcu, P. A. van Aken, B. Keimer, M. Isobe, and M. Hepting, *Science Advances* **7**, eabl8091 (2021).
- ³⁰ B.-X. Wang, H. Zheng, E. Krivyakina, O. Chmaissem, P. P. Lopes, J. W. Lynn, L. C. Gallington, Y. Ren, S. Rosenkranz, J. F. Mitchell, and D. Phelan, *Physical Review Materials* **4**, 084409 (2020).
- ³¹ Q. Li, C. He, J. Si, X. Zhu, Y. Zhang, and H.-H. Wen, *Communications Materials* **1**, 16 (2020).
- ³² C. He, X. Ming, Q. Li, X. Zhu, J. Si, and H.-H. Wen, *Journal of Physics: Condensed Matter* **33**, 265701 (2021).
- ³³ M. Huo, Z. Liu, H. Sun, L. Li, H. Lui, C. Huang, F. Liang, B. Shen, and M. Wang, *Chinese Physics B* (2022).
- ³⁴ V. V. Poltavets, K. A. Lokshin, S. Dikmen, M. Croft, T. Egami, and M. Greenblatt, *Journal of the American Chemical Society* **128**, 9050 (2006).

- ³⁵ V. V. Poltavets, K. A. Lokshin, M. Croft, T. K. Mandal, T. Egami, and M. Greenblatt, *Inorganic Chemistry* **46**, 10887 (2007).
- ³⁶ V. V. Poltavets, M. Greenblatt, G. H. Fecher, and C. Felser, *Physical Review Letters* **102**, 046405 (2008).
- ³⁷ V. V. Poltavets, K. A. Lokshin, A. H. Nevidomskyy, M. Croft, T. A. Tyson, J. Hadermann, G. V. Tendeloo, T. Egami, G. Kotliar, N. ApRoberts-Warren, A. P. Dioguardi, N. J. Curro, and M. Greenblatt, *Physical Review Letters* **104**, 206403 (2010).
- ³⁸ Y. Sakurai, N. Chiba, Y. Kimishima, and M. Uehara, *Physica C: Superconductivity* **487**, 27 (2013).
- ³⁹ J. Hao, X. Fan, Q. Li, X. Zhou, C. He, Y. Dai, B. Xu, X. Zhu, and H.-H. Wen, *Physical Review B* **103**, 205120 (2021), 2101.02979.
- ⁴⁰ Q. Li, C. He, X. Zhu, J. Si, X. Fan, and H.-H. Wen, *Science China Physics, Mechanics & Astronomy* **64**, 227411 (2021).
- ⁴¹ A. Sharma, B. Devanarayanan, P. D. Patel, and N. Singh, (2022), 10.48550/arxiv.2206.07539.
- ⁴² E. Breckenfeld, Z. Chen, A. R. Damodaran, and L. W. Martin, *ACS Applied Materials & Interfaces* **6**, 22436 (2014).
- ⁴³ J. W. Freeland, M. Van Veenendaal, and J. Chakhalian, *Journal of Electron Spectroscopy and Related Phenomena* **208**, 56 (2016).
- ⁴⁴ I.-C. Tung, G. Luo, J. H. Lee, S. H. Chang, J. Moyer, H. Hong, M. J. Bedzyk, H. Zhou, D. Morgan, D. D. Fong, *et al.*, *Physical Review Materials* **1**, 053404 (2017).
- ⁴⁵ T. Kim, T. Paudel, R. Green, K. Song, H.-S. Lee, S.-Y. Choi, J. Irwin, B. Noesges, L. Brillson, M. Rzchowski, *et al.*, *Physical Review B* **101**, 121105 (2020).
- ⁴⁶ A. S. Disa, D. P. Kumah, A. Malashevich, H. Chen, D. A. Arena, E. D. Specht, S. Ismail-Beigi, F. Walker, and C. H. Ahn, *Physical review letters* **114**, 026801 (2015).
- ⁴⁷ B. H. Goodge, D. Li, K. Lee, M. Osada, B. Y. Wang, G. A. Sawatzky, H. Y. Hwang, and L. F. Kourkoutis, *Proceedings of the National Academy of Sciences* **118**, e2007683118 (2021).
- ⁴⁸ M. Rossi, H. Lu, A. Nag, D. Li, M. Osada, K. Lee, B. Y. Wang, S. Agrestini, M. Garcia-Fernandez, J. Kas, *et al.*, *Physical Review B* **104**, L220505 (2021).

REVIEWERS' COMMENTS

Reviewer #2 (Remarks to the Author):

In the revised manuscript, the authors have addressed the questions/comments properly. I would recommend publication in Nat. Commun.

Reviewer #3 (Remarks to the Author):

The authors have done an impressive job of addressing the critiques of the Referees and revising the manuscript accordingly. I recommend that the revised manuscript be published in Nature Communications.

**Response to reviewers for “Limits to the strain engineering of
layered square-planar nickelate thin films” (NCOMMS-22-29067)**

(Dated: February 10, 2023)

REVIEWER #2 (REMARKS TO THE AUTHOR):

In the revised manuscript, the authors have addressed the questions/comments properly. I would recommend publication in Nat. Commun.

Our response:

We appreciate the reviewer's consideration of our work and thank them for helping us improve our manuscript.

REVIEWER #3 (REMARKS TO THE AUTHOR):

The authors have done an impressive job of addressing the critiques of the Referees and revising the manuscript accordingly. I recommend that the revised manuscript be published in Nature Communications.

Our response:

We appreciate the reviewer's consideration of our work and thank them for helping us improve our manuscript.